# Predicting individual patient and hospital-level discharge using machine learning
Jia Wei [1], Jiandong Zhou [1], Zizheng Zhang[2], Kevin Yuan [2], Qingze Gu[1], Augustine Luk[1], Andrew J. Brent[1,3], David A. Clifton [4,5], A. Sarah Walker[1,6,7] & David W. Eyre [2,3,6,7] ✉

## Abstract

**Background** Accurately predicting hospital discharge events could help improve patient flow and the efficiency of healthcare delivery. However, using machine learning and diverse electronic health record (EHR) data for this task remains incompletely explored.
**Methods** We used EHR data from February-2017 to January-2020 from Oxfordshire, UK to predict hospital discharges in the next 24 h. We fitted separate extreme gradient boosting models for elective and emergency admissions, trained on the first two years of data and tested on the final year of data. We examined individual-level and hospital-level model performance and evaluated the impact of training data size and recency, prediction time, and performance in subgroups.
**Results** Our models achieve AUROCs of 0.87 and 0.86, AUPRCs of 0.66 and 0.64, and F1 scores of 0.61 and 0.59 for elective and emergency admissions, respectively. These models outperform a logistic regression model using the same features and are substantially better than a baseline logistic regression model with more limited features. Notably, the relative performance increase from adding additional features is greater than the increase from using a sophisticated model. Aggregating individual probabilities, daily total discharge estimates are accurate with mean absolute errors of 8.9% (elective) and 4.9% (emergency). The most informative predictors include antibiotic prescriptions, medications, and hospital capacity factors. Performance remains robust across patient subgroups and different training strategies, but is lower in patients with longer admissions and those who died in hospital.
**Conclusions** Our findings highlight the potential of machine learning in optimising hospital patient flow and facilitating patient care and recovery.

## Plain language summary

Predicting when hospital patients are ready to be discharged could help hospitals run more smoothly and improve patient care. In this study, we used three years of patient records from Oxfordshire, UK, to build a machine learning model that predicts discharges within the next 24 h. Our model includes both planned and emergency admissions. The model performs well at accurately predicting the probability, or chance, that an individual patient will be discharged and also estimating the total number of discharges each day. Important information for making the predictions includes whether patients are taking antibiotics and other medications, and whether the hospital is crowded. Overall, we show that machine learning could help hospitals manage patient flow and improve patient care.

Increasing demand for healthcare, driven by changing population demographics, a rise in the prevalence of chronic diseases, societal changes, and technological advances, places significant strain on hospital resources[1]. In the United Kingdom (UK), the National Health Service (NHS) has faced escalating demand pressures in recent years, with an increasing number of admissions, prolonged waiting times in Emergency Departments, and financial challenges[2,3]. This has been exacerbated by the COVID-19 pandemic, resulting in substantial backlogs in both urgent and routine care[4]. With healthcare resources being inherently limited, there is a pressing need

to enhance the efficiency of healthcare services and improve hospital capacity management. A critical component is patient flow within hospitals, referring to the movement of patients from admission to discharge while ensuring they receive appropriate care and resources[5]. Optimising this could improve patient experiences, avoid delays in treatment, improve health outcomes, and reduce costs[6].

Accurately predicting when patients will be discharged from the hospital could improve patient flow, e.g., prompting clinicians when patients are approaching readiness for discharge, facilitating booking transport

---

[1]Nuffield Department of Medicine, University of Oxford, Oxford, UK. [2]Big Data Institute, Nuffield Department of Population Health, University of Oxford, Oxford, UK. [3]Oxford University Hospitals NHS Foundation Trust, Oxford, UK. [4]Department of Engineering Science, University of Oxford, Oxford, UK. [5]OSCAR (Oxford Suzhou Centre for Advanced Research), University of Oxford, Suzhou, Jiangsu, China. [6]The National Institute for Health Research Health Protection Research Unit in Healthcare Associated Infections and Antimicrobial Resistance at the University of Oxford, Oxford, UK. [7]National Institute for Health Research Oxford Biomedical Research Centre, Oxford, UK. ✉e-mail: david.eyre@bdi.ox.ac.uk

home, enabling timely preparation of discharge medication and documentation, and pre-emptively arranging room cleaning. Currently, discharge predictions are made in most hospitals at the individual patient level by clinical teams based on the patient's diagnosis and status and are updated throughout their hospital stay. However, these assessments can be subjective and variable and may not always be captured in electronic healthcare record (EHR) systems, posing challenges to efficient operational management. Therefore, there is growing interest in leveraging automated prediction models to forecast the length of stay (LOS) and discharge timing, both individually and hospital-wide.

Discharge prediction has therefore become a key target for clinical machine learning researchers. Several previous studies have attempted to predict discharge within a fixed time window (Table 1), with studies typically predicting discharge within the next 24, 48, or 72 h. Some of these studies have focused on specific patient populations, e.g., surgical patients[7,8] or those with cardiovascular disease[9], while others predict discharge for whole hospitals[10–14]. A range of different classical machine learning approaches have been evaluated, including random forests, gradient boosted trees, and multilayer perceptron neural networks. Input features being considered typically include details relating to the index date the prediction is being made on, past medical history, prior length of stay, demographics, current vital signs and laboratory markers, diagnoses, procedures, and medications (Supplementary Table 1). Performance is typically modest in most models, although some perform better, with area under the receiver operating curve (AUROC) values ranging from 0.70 to 0.86 in whole hospital models. One notable exception is a model that also included data on EHR-user interactions[10], such as the frequency of clinical notes being updated, viewed, or printed, which achieved an AUROC of 0.92 for predicting discharge within 24 h. However, this study only included the first admission for each patient, potentially under-representing complex patients who may be frequently re-admitted. It also excluded patients who died during hospitalisation. Additionally, in most previous studies several key areas relevant to implementation were not fully explored, including the impact of training data size and recency, the time of day or day during the week a prediction is made, and performance relative to length of stay and specific patient subgroups, including different sociodemographic groups and those affected by health inequalities. Apart from two studies[11,13], all of the aforementioned studies either evaluated individual-level discharge prediction performance or hospital-wide predictions, but did not combine the two in a single approach.

In this study, we used diverse EHR-derived features and data from a large UK teaching hospital group to develop machine learning models to predict individual patient-level hospital discharge within the next 24 h. By aggregating individual predictions, our models were also successful in predicting the total hospital-wide number of discharges expected.

## Methods
### Data and setting
We used data from the Infections in Oxfordshire Research Database (IORD) which contains deidentified electronic healthcare records from Oxford University Hospitals (OUH) NHS Foundation Trust. OUH consists of four teaching hospitals in Oxfordshire, United Kingdom, with a total of ~1100 beds, serving a population of ~755,000 people and providing specialist services to the surrounding region.

We extracted data for all adult inpatients ( ≥ 16 years) from 01 February 2017 to 31 January 2020 who had an ordinary admission based on NHS patient classification codes (i.e., excluding day case, regular day admission, and regular night admissions, because the expected length of stay was known a priori for these patients). We excluded patients whose admission specialty was obstetrics or paediatrics, as these specialties used a different EHR system and/or discharge pathway. We grouped admissions into elective admissions (those scheduled in advance) and emergency admissions (those who entered hospital through the Emergency Department or other emergency admissions units) based on admission method codes (Supplementary Fig. 1).

### Feature selection and data pre-processing
Domain knowledge and prior literature were used to determine which features within the dataset were potentially informative for predicting patient discharge. A total of 1152 features were created and grouped into 16 feature categories (Supplementary Methods), including index date-related features, demographics, comorbidities, current admission, ward stay, current diagnostic category, procedures, antibiotics prescriptions, medication, microbiology tests, radiology investigation, readmissions and previous hospital stay, hospital capacity factors reflecting crowdedness, vital signs, and laboratory tests.

To produce features summarising expected length of stay (LOS) by each patient's primary diagnosis, we first summarised all primary diagnostic ICD-10 codes using Summary Hospital-level Mortality Indicator (SHMI) diagnosis groupings[15]. Using data from all hospital admissions within the training dataset (01 February 2017 to 31 January 2019), we calculated as features the mean, standard deviation, median, maximum, and minimum of the LOS for each diagnostic category, to capture the effects of a current diagnostic category on future discharge probability[16]. We only used the training data to calculate the LOS characteristics of each diagnosis category, even when applying these estimates to the test dataset, to avoid possible data leakage, i.e., avoiding revealing information to the model that gives it an unrealistic advantage to make better predictions. ICD10 codes were assigned at discharge but were used as a proxy for the clinician's working diagnosis (not available in our dataset) to inform model predictions in real time.

For vital signs and laboratory tests, we used both numerical values reflecting the measurements themselves and the number of measurements within a particular time window, reflecting the fact that the decision to measure a vital sign or laboratory test is potentially informative in addition to the actual result obtained[17]. For example, clinicians may order additional laboratory measurements or record vital signs more frequently if patients are unstable[18]. To reduce collinearity, we grouped the number of measurements for vital signs (heart rate, respiratory rate, systolic blood pressure, diastolic blood pressure, temperature, oxygen saturation, O2 L/min, O2 delivery device, AVPU score, NEWS2 score), full blood counts (haemoglobin, haematocrit, mean cell volume, white cell count, platelets, neutrophils, lymphocytes, eosinophils, monocytes, basophils), renal function (creatinine, urea, potassium, sodium, estimated glomerular filtration rate (eGFR)), liver function (alkaline phosphatase, aspartate aminotransferase, alanine transaminase, albumin, bilirubin), bone profiles (adjusted calcium, magnesium, phosphate), clotting (activated partial thromboplastin time, prothrombin time), blood gases (arterial, venous, or capillary combined, as labelling of blood gas type was not always complete/reliable) (base excess, partial pressure of oxygen, partial pressure of carbon dioxide, lactate, arterial blood pH), and lipids (triglycerides, high-density lipoprotein cholesterol, total cholesterol, low-density lipoprotein cholesterol), respectively. The number of measurements for other blood tests were included individually.

We pre-processed the data by setting to missing implausible extreme values (Supplementary Data 7) not compatible with life, which typically represented uncorrectable errors in data entry, e.g., height 10 m, temperature 20 °C. Categorical features were one-hot encoded. We did not perform data truncation and standardisation as decision trees-based algorithms are insensitive to the scale of the features, and we did not perform imputation of missing values because extreme gradient boosting (XGB) models can handle missing values by default using a 'missing incorporated in attribute' algorithm, which treats missing values as a separate category[19,20]. The proportion of missingness for features ranged from 0 to 88%, with the values of less commonly obtained laboratory tests exhibiting the greatest proportions of missingness (Supplementary Fig. 2).

### Prediction tasks
We predicted hospital discharge events within 24 h of an index date and time for all patients currently in the hospital (individual-level predictions). Model prediction probabilities were also aggregated to predict the total number of patients currently in hospital across the 4 hospitals within the

**Table 1 | Previous studies predicting discharge within a fixed time window**

| Reference | Population | Outcome | Model | Performance, AUROC where available |
|---|---|---|---|---|
| 7 | Adult patients discharged from inpatient surgical care in the US from May 1, 2016, to August 31, 2017; 15,201 hospital discharges | Discharge within 24 h | Multilayer perceptron neural network | AUROC 0.84 |
| 8 | Adult surgical patients discharged from inpatient care between July 2018 and February 2020; 10,904 patients during 12,493 inpatient visits | Discharge within 48 h | RF | AUROC 0.81 |
| 9 | Inpatients with cardiovascular diseases admitted to Asan Medical Centre in Korea between 2000 to 2016; 669,667 records | Discharge within the next 72 h (predictions not made on the day of discharge) | Extreme gradient boosting (XGB) | AUROC 0.87 |
| 10 | Adult patients admitted to Vanderbilt University Medical Centre in 2019; 26,283 patients | Discharge within 24 h | Light gradient boosting machine (LGBM) | AUROC 0.92 with user-EHR interactions; AUROC 0.86 without user-EHR interactions. |
| 11 | Patients admitted to a mid-Atlantic academic medical centre from 2011–2013; 8852 patient visits and 20,243 individual patient days | Discharge within 7 and 17 h (from 7 am) | Logistic regression (LR); Random Forest (RF) | Sensitivity: LR: 65.9; RF: 60.0; Specificity: LR: 52.8; RF: 66.0 |
| 12 | Adult patients admitted to a community hospital in Maryland, USA between April 2016 and August 2019; 120,780 discharges for 12,470 patients | Discharge on the same day, by the next day, within the next 2 days | RF | AUROC 0.80 (same day); AUROC 0.70 (next day) |
| 13 | Inpatients admitted at Beth Israel Deaconess Medical Centre between January 2017 and August 2018; 63,432 unique admissions (41,726 unique patients) | Discharge within 1 day, discharge within 2 days | LR, CART decision trees, Optimal trees, RF, Gradient boosted trees | Discharge within 1 day, AUROC 0.84; Discharge within 2 days, AUROC 0.82 |
| 14 | Patient encounters from 14 different Kaiser Permanente facilities in northern California from November 1, 2015 to December 31, 2017; 910,366 patient-days across 243,696 patients hospitalisations | Discharge within 1 day | LR, Lasso, RF, GBM | GBM, AUROC 0.73 |

We searched Google Scholar and PubMed for studies up to 30 April 2024, using the search terms 'machine learning' AND ('hospital discharge prediction', OR 'patient flow'). AUROC: area under the receiver operating curve. XGB: Extreme gradient boosting. GBM: gradient boosting machine. LR: logistic regression. RF: random forest. CART: classification and regression tree. The features used in each model are summarised in Supplementary Table 1.

OUH hospital group who would be discharged within the next 24 h following the index date time (hospital-level predictions). We used a pragmatic and operationally relevant endpoint, predicting discharges from hospital with any outcome (discharged alive, died, or transferred to another hospital outside of OUH [2% of discharges]).

Each patient contributed once to the dataset per day during their hospital admission. Separate models were built for emergency and elective admissions as predictive features may be different depending on the reason for admission. For the main analyses, predictions were made at 12 pm for both the training and test data, i.e., to mirror the approximate time that most hospital ward rounds are completed by (accepting some may be completed earlier). The probability of a patient being discharged by 12 pm on the following day was obtained. However, in real-world use model predictions are likely to be applied throughout the day as new patients are admitted and others discharged. We therefore performed three sets of sensitivity analyses (all predicting discharges within the next 24 h): 1) train and predict discharge at other times of day (midnight, 6 am, 6 pm); 2) train the model and predict discharge using data drawn randomly throughout the day, using data available at 2 hourly intervals, i.e., midnight, 2 am, 4 am, …, 10 pm); 3) train and predict discharge at different times of day (e.g., train at 12 pm, test at 6 am).

We used extreme gradient boosting (XGB) models to predict discharge within the next 24 h from an index date. Gradient-boosted trees are an additive method iteratively combining fitted decision trees that identify and set cut-off points by splitting the values of input features into those associated with an outcome of interest (conceptually similar to the model deciding what constitutes a 'normal' vs. 'abnormal' value). Each individual tree has relatively weak predictive performance, but these are combined into a single high-performance ensemble model after proportionally weighting individual tree contributions[21]. Each fitted decision tree (base learner with low complexity) targets the prediction residuals of the preceding tree. That is, at each step a new decision tree is built, targeting the fraction of the output not well explained by the current tree model. In this way, a new tree model is continuously added to the current collection to correct the mistakes made by the previous one. Such a sequential training approach in gradient-boosted trees is different from the independent training approach in random forests. Models were trained using data from the first two years of the study (01 February 2017 to 31 January 2019) and evaluated using data from the final year of the study (01 February 2019 to 31 January 2020). To examine whether the trained model could be used to predict hospital discharge following the COVID-19 pandemic, we also used data from 01 February 2021 to 31 January 2022 as an additional held-out test dataset.

We randomly split the training data, using 80% of the data for the main model training. Within this, a Bayesian optimisation for hyperparameters was performed by employing Tree-based Parzen estimators (TPE) to search through a wide potential hyperparameter space, maximising the AUROC during 5-fold cross-validation. Hyperparameters tuned included learning rate (learning_rate), the number of trees (n_estimators), the fraction of features to use (colsample_bytree), the fraction of observations to subsample at each step (subsample), maximum number of nodes allowed from the root to the farthest leaf of a tree (max_depth), minimum weight required to create a new node in a tree (min_child_weight), the minimum loss reduction required to make a split (gamma), L1 regularisation term on weight (reg_alpha), and L2 regularisation term on weight (reg_lambda). Unlike grid and random searches, which independently tune hyperparameters, Bayesian hyperparameter optimization is an informed search algorithm that each iteration learns from previous iterations and combines this with a prior distribution to update the posterior of the optimization function[22]. Hyperparameter optimisation was undertaken separately for each model fitted. The final hyperparameter choices for the main models are shown in Supplementary Table 2.

We used the built-in scale_pos_weight in the XGB classifier to account for class imbalance, i.e. the fact there are more non-discharge events than discharges. No imputation of missing data was performed, as XGB can handle missing data by design. We used the remaining 20% of the training data as validation data for feature selection, calibrating the predicted probability from XGB models using isotonic regression[23,24] and determining the best threshold for predicting an individual patient discharge event by optimising the F1 score (jointly maximising precision and recall). Initially, all 1152 features were used to fit each model. Models were then re-fitted with progressively fewer features, retaining the top-ranked features from each full model. Performance in the validation data and training time was used to select the optimal number and list of input features for the main analyses. The model pipeline is shown in Fig. 1. Model performance was compared to two baseline logistic regression (LR) models: 1) with fewer selected features (age, sex, day of the week, hours since admission), for benchmarking against a relatively simple model; 2) with the same set of features as the final XGB models, using L2 regularization, and with missing data imputed using median values for continuous features and the mode for categorical features.

For hospital-level prediction, we calculated the total number of predicted discharges expected in the next 24 h across all elective or emergency admissions in the 4 OUH hospitals by summing the individual-level predicted discharge probabilities after calibration.

## Performance assessment

Individual-level model performance was evaluated using sensitivity (recall), specificity, balanced accuracy (arithmetic mean of sensitivity and specificity), positive predictive value (PPV, precision), negative predictive value (NPV), F1 score (harmonic mean of precision and recall), AUROC, and area under the precision-recall curve (AUPRC). PPV and NPV provide interpretable real-world metrics relating to actual discharge decision making and prediction performance. Although they depend on the prevalence of discharge events, our estimates are likely to apply to hospitals with similar daily discharge rates. F1 scores and AUPRC are also impacted by prevalence, and for a given sensitivity and specificity will be lower as prevalence falls, which should be considered when comparing subgroups with different discharge prevalence.

For hospital-level prediction, we summarised the accuracy of predictions of the total number of patients discharged using normalised mean absolute error (MAE, %), i.e. the mean of the differences in predicted and actual discharges each day (over the 365 predictions in the test dataset) divided by the mean number of discharges per day.

We calculated the SHapley Additive exPlanations (SHAP) values[25] for each feature in the training dataset to determine feature importance.

We examined the model performance in different subgroups by age, sex, ethnicity, index of multiple deprivation score, weekday of the index date, admission specialty, comorbidity score, source of admission, days since admission, and discharge outcome (alive or died). We also used equalised odds differences to compare model fairness, by checking if either the per subgroup true positive rate or true negative rate differed from the overall rate by greater than an illustrative threshold of 0.1[26].

We also evaluated model performance using the same test data, but with different lengths of training data from 1 to 24 months, and the impact of training data recency using 12 months of training data at varying time intervals before the fixed test dataset.

## Statistics and reproducibility

Model performance was summarised using percentages. No statistical tests were conducted.

Data processing and analyses were performed in Python 3.11 using the following packages: numpy (version 1.26.4), pandas (version 2.2.0), scipy (version 1.12.0), scikit-learn (version 1.4.1), xgboost (version 2.0.3), hyperopt (version 0.2.7), and in R (version 4.3.2) using the following packages: cowplot (version 1.1.1), timeDate (version 4021.104), ggplot2 (version 3.4.4), and tidyverse (version 1.3.2).

## Ethics committee approval

Deidentified data were obtained from the Infections in Oxfordshire Research Database which has approvals from the National Research Ethics Service South Central – Oxford C Research Ethics Committee (19/SC/

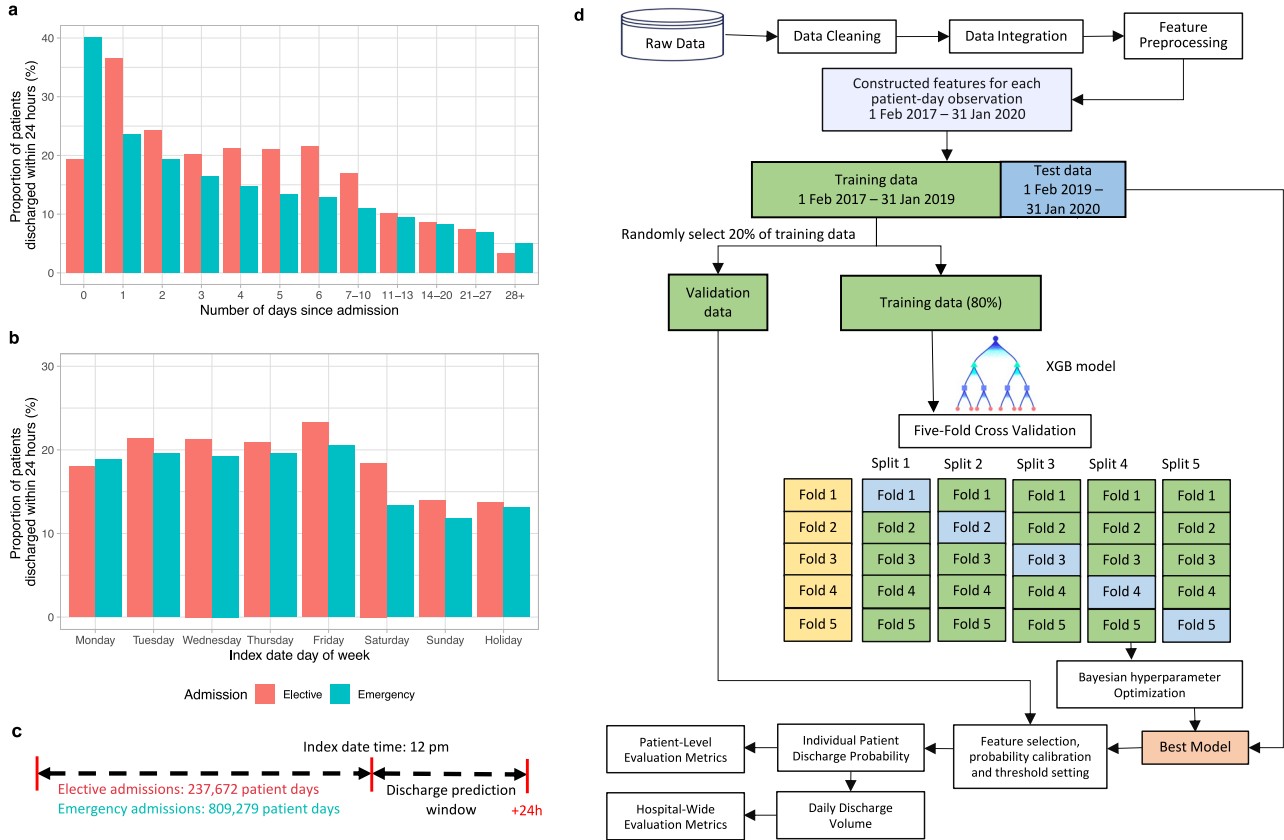

**Fig. 1 | Overview of model development. a** The proportion of patients discharged from hospital within the next 24 h in elective and emergency admissions by number of days since admission. **b** The proportion of patients discharged from hospital within the next 24 h in elective and emergency admissions by day of week of the index date. **c** Diagram of the prediction problem. The binary prediction problem was defined by classifying the outcome as 'positive' (discharge occurred within the next 24 h) or 'negative' (discharge did not occur within the next 24 h) separately for elective and emergency admissions. Predictions were made at 12 pm. **d** Analysis

pipeline for the prediction of hospital discharge within the next 24 h. Extreme gradient boosting (XGB) models were trained on the extracted labels and features from admissions between 01 February 2017 to 31 January 2019, and was tested on admissions between 01 February 2019 and 31 January 2020. Five-fold cross validation was used for hyperparameter tuning, and 20% randomly selected validation data was used for feature selection, probability calibration, and threshold setting. The best model was used to predict hospital discharges in the test data, and model performance was examined.

0403), the Health Research Authority and the National Confidentiality Advisory Group (19/CAG/0144), including provision for use of pseudonymised routinely collected data without individual patient consent. Patients who choose to opt out of their data being used in research are not included in the study. The study was carried out in accordance with all relevant guidelines and regulations.

## Results

From 01 February 2017 to 31 January 2020, 52,590 elective admissions and 202,633 emergency admissions were recorded. Using 12 pm as the prediction time, 63,909 (25.0%) short admissions were excluded from the main analyses because these admissions did not include time in hospital at 12 pm (Supplementary Fig. 1), i.e. some admissions started after 12 pm and these patients were discharged before 12 pm the following day. All other admissions < 24 h but spanning 12 pm were included. This resulted in a total of 48,039 elective admissions (38,627 patients) and 143,275 emergency admissions (86,059 patients) included in the analyses. The median (IQR) age at admission was 65 (47–79) years, 97,869 (51.2%) patients were female, and 148,060 (77.4%) were recorded as being of white ethnicity (ethnicity missing in 33,626, 17.6%). The length of hospital stay was right-skewed, with most patients discharged within a few days, but some staying considerably longer. The median (IQR) length of stay was 2.2 (1.2, 5.0) days for elective admissions and 2.1 (0.8, 6.0) days for emergency admissions (Supplementary Fig. 3). The distribution of demographic characteristics was similar between the training and test datasets (Table 2).

### Model performance for individual-level prediction

Predicting discharge at 12 pm, 237,672 and 809,279 patient days for elective and emergency admissions were included in the analyses. 47,177 (19.8%) and 141,531 (17.5%) discharge events within 24 h of the index date were observed, respectively, reflecting the daily discharge rate. The mean proportion of patients discharged from the hospital within the next 24 h decreased as the prior length of stay in the current admission increased, and varied between emergency and elective admissions, and the day of the week of the index date (Fig. 1).

Using the validation dataset, we evaluated model performance with varying numbers of features, ranging from 10 to all 1152 features. Model performance initially improved as more features were included, but then plateaued with ≥200 features (and for some metrics even slightly declined) (Supplementary Fig. 4). As expected, training time increased with the number of features (Supplementary Fig. 4). We therefore used the top 200 ranked emergency and elective model features in all subsequent emergency and elective models.

For elective admissions, the AUROC for predicting discharge within 24 h was 0.871, and the AUPRC was 0.658 (Supplementary Fig. 5). Using a probability threshold that optimised F1 score in the validation dataset, the PPV was 0.555, NPV 0.911, and F1 score 0.609 (Table 3). The performance for emergency admissions was slightly lower than elective admissions, with an AUROC of 0.860, AUPRC 0.644 (Supplementary Fig. 5), PPV 0.571, NPV 0.912, and F1 score 0.593 (Table 3). Predicted probabilities reflected the real probabilities of discharge after calibration, with calibration errors of

**Table 2 | Baseline characteristics of 48,039 elective and 143,275 emergency admissions between 01 February 2017 to 31 January 2020 used in training and testing discharge prediction models**

| | Elective (N = 48,039) | | Emergency (N = 143,275) | | Total (N = 191,314) |
|---|---|---|---|---|---|
| | Training (N = 32,832) | Test (N = 15,207) | Training (N = 92,611) | Test (N = 50,664) | |
| **Age (years)** | | | | | |
| Median (Q1, Q3) | 62.0 (48.0, 73.0) | 62.0 (48.0, 73.0) | 67.0 (47.0, 81.0) | 66.0 (46.0, 81.0) | 65.0 (47.0, 79.0) |
| **Sex** | | | | | |
| Female | 16,409 (50.0%) | 7603 (50.0%) | 47,790 (51.6%) | 26,067 (51.5%) | 97,869 (51.2%) |
| Male | 16,423 (50.0%) | 7604 (50.0%) | 44,821 (48.4%) | 24,597 (48.5%) | 93,445 (48.8%) |
| **Ethnicity** | | | | | |
| White | 23,977 (73.0%) | 10,961 (72.1%) | 73,611 (79.5%) | 39,511 (78.0%) | 148,060 (77.4%) |
| Mixed | 173 (0.5%) | 92 (0.6%) | 627 (0.7%) | 390 (0.8%) | 1282 (0.7%) |
| Asian | 712 (2.2%) | 402 (2.6%) | 2198 (2.4%) | 1264 (2.5%) | 4576 (2.4%) |
| Black | 355 (1.1%) | 185 (1.2%) | 1040 (1.1%) | 566 (1.1%) | 2146 (1.1%) |
| Other | 271 (0.8%) | 155 (1.0%) | 792 (0.9%) | 406 (0.8%) | 1624 (0.8%) |
| Unknown | 7344 (22.4%) | 3412 (22.4%) | 14,343 (15.5%) | 8527 (16.8%) | 33,626 (17.6%) |
| **IMD deprivation score** | | | | | |
| Median (Q1, Q3) | 9.8 (5.9, 15.5) | 9.9 (6.2, 15.8) | 10.5 (6.2, 16.5) | 10.3 (6.5, 16.3) | 10.2 (6.2, 16.1) |
| Missing | 525 | 212 | 1122 | 555 | 2414 |
| **Admission source** | | | | | |
| Usual place of residence | 32,232 (98.2%) | 14,959 (98.5%) | 87,493 (94.6%) | 48,212 (95.4%) | 182,896 (95.7%) |
| Other hospital provider | 487 (1.5%) | 196 (1.3%) | 4210 (4.6%) | 2023 (4.0%) | 6916 (3.6%) |
| Other | 92 (0.3%) | 36 (0.2%) | 758 (0.8%) | 324 (0.6%) | 1210 (0.6%) |
| Missing | 21 | 16 | 150 | 105 | 292 |
| **Length of stay (days)** | | | | | |
| Median (Q1, Q3) | 2.2 (1.2, 5.1) | 2.2 (1.2, 4.9) | 2.1 (0.9, 6.1) | 2.0 (0.8, 5.7) | 2.1 (1.0, 5.6) |

*IMD:* Index of multiple deprivation (higher scores indicate greater deprivation, range 0–73.5).

0.003 and 0.001 for elective and emergency admissions, respectively (Supplementary Fig. 6). Performance in training, validation, and test data is shown in Supplementary Table 3.

The XGB models showed substantially better performance than the simple baseline LR models, which had AUROCs of 0.629, 0.708, AUPRCs of 0.269, 0.349, and F1 scores of 0.385, 0.409 for elective and emergency admissions, respectively. LR models with the same 200 features as the XGB models outperformed the simple LR models but performed worse than the XGB models. The performance difference was larger between the baseline and 200 feature LR models (e.g. AUROC 0.821 vs 0.629 for elective patients) than that between the LR models with 200 features and XGB models (AUROC 0.821 vs 0.871 for elective patients). Hence, including additional appropriately chosen features improved model performance by more than using a more sophisticated modelling technique, but with both approaches adding to performance (Table 3). When combining elective and emergency admissions into a single XGB model, performance was similar to that of the XGB model for emergency admissions, with an AUROC of 0.859, AUPRC 0.634, PPV 0.561, NPV 0.909, and F1 score 0.587.

When we used the trained model to predict discharge in the post-COVID test data (01 February 2021 to 31 January 2022), performance remained similar for elective admissions, with an AUROC of 0.864 and AUPRC of 0.614, but was lower for emergency admissions, with an AUROC of 0.820 and AUPRC of 0.543 (Table 3).

**Model performance for hospital-level prediction**
Summing individual-level predicted discharge probabilities, the predicted total number of discharges across the 4 OUH hospitals accurately reflected daily fluctuations in discharge numbers during the week for elective and emergency patients, and the performance was similar across the whole test

data period (Fig. 2). The MAE was 8.9% (MAE = 3.7 discharges, mean total discharges = 41) for elective admissions using the XGB model, lower than the 10.7% (MAE = 4.4/41) obtained using baseline LR models. Errors were well controlled on most days, IQRs for absolute errors (normalised by the actual number of discharges) for baseline LR and XGB models were 5.9–18.4% and 4.9–15.8% respectively (Supplementary Fig. 7). For emergency admissions MAEs were 4.9% (MAE = 7.2/146; IQR 2.3–9.3%) using XGB compared with 5.8% (MAE = 8.6/146; IQR 2.8–11.4%) using the baseline LR models (Table 3, Supplementary Fig. 7). MAEs were higher in post-COVID test data, using XGB these were 11.6% (MAE = 3.9/34) for elective (higher in percentage terms in part because of lower total discharge numbers), and 10.0% (MAE = 15.0/150) for emergency patients (Table 3, Supplementary Fig. 8).

**Subgroup performance and model fairness**
Using balanced accuracy to jointly summarise sensitivity and specificity, model performance was broadly similar by sex, ethnicity, and deprivation score, but some variations existed in other subgroups. For elective admissions, predictions made on Monday and Sunday, and for patients admitted to trauma and orthopaedics or acute medicine had lower balanced accuracy. For emergency admissions, balanced accuracy was lower on Sundays, and for patients admitted to trauma and orthopaedics and medical subspecialties. For both elective and emergency admissions, balanced accuracy was lower in those > 80 years, those with high comorbidity scores, with increasing days since admission, and in admissions from other hospital providers. Balanced accuracy was also substantially lower in those who died in hospital than those who were discharged alive (with lower AUROCs, AUROC = 0.712 vs 0.870, 0.762 vs 0.862 for elective and emergency admissions, respectively). In addition to lower accuracy, lower discharge

**Table 3 | Model performance of extreme gradient boosting (XGB) models with 200 features, baseline logistic regression (LR) model, and LR model with 200 features predicting 24-hour discharge in the test dataset (01 February 2019 to 31 January 2020) and an additional test dataset post-COVID (01 February 2021 to 31 January 2022)**

| Admission type | Model | Accuracy | Balanced accuracy | Sensitivity/Recall | Specificity | PPV/Precision | NPV | F1-score | AUROC | AUPRC | MAE (%) |
|---|---|---|---|---|---|---|---|---|---|---|---|
| Test data: 01/02/2019 to 31/01/2020 | | | | | | | | | | | |
| Elective | XGB (200 features) | 0.823 | 0.767 | 0.673 | 0.861 | 0.555 | 0.911 | 0.609 | 0.871 | 0.658 | 8.9 |
| Elective | LR (baseline) | 0.464 | 0.596 | 0.820 | 0.372 | 0.252 | 0.889 | 0.385 | 0.629 | 0.269 | 10.7 |
| Elective | LR (200 features) | 0.696 | 0.740 | 0.815 | 0.666 | 0.386 | 0.933 | 0.524 | 0.821 | 0.538 | 10.6 |
| Emergency | XGB (200 features) | 0.844 | 0.756 | 0.616 | 0.896 | 0.571 | 0.912 | 0.593 | 0.860 | 0.644 | 4.9 |
| Emergency | LR (baseline) | 0.637 | 0.654 | 0.682 | 0.626 | 0.292 | 0.897 | 0.409 | 0.708 | 0.349 | 5.8 |
| Emergency | LR (200 features) | 0.718 | 0.738 | 0.769 | 0.707 | 0.372 | 0.931 | 0.501 | 0.813 | 0.507 | 5.6 |
| Overall | XGB (200 features) | 0.837 | 0.752 | 0.615 | 0.888 | 0.561 | 0.909 | 0.587 | 0.859 | 0.634 | 4.6 |
| Overall | LR (baseline) | 0.589 | 0.642 | 0.726 | 0.558 | 0.276 | 0.898 | 0.400 | 0.694 | 0.327 | 5.4 |
| Overall | LR (200 features) | 0.700 | 0.733 | 0.787 | 0.680 | 0.363 | 0.932 | 0.497 | 0.809 | 0.497 | 5.0 |
| Test data: 01/02/2021 to 31/01/2022 | | | | | | | | | | | |
| Elective | XGB (200 features) | 0.825 | 0.753 | 0.638 | 0.869 | 0.532 | 0.911 | 0.580 | 0.864 | 0.614 | 11.6 |
| Emergency | XGB (200 features) | 0.835 | 0.703 | 0.501 | 0.906 | 0.528 | 0.896 | 0.514 | 0.820 | 0.543 | 10.0 |

The baseline LR models only included age, sex, day of the week, and hours since admission. *PPV*: Positive predictive value, *NPV*: Negative predictive value, *AUROC*: Area under the receiver operating curve, *AUPRC*: Area under the precision-recall curve, *MAE*: Normalised mean absolute error (mean difference in predicted and actual discharges per day divided by the mean number of discharges per day).

rates in some subgroups including with increasing prior length of stay, also contributed to lower in PPV in those groups, albeit with linked increases in NPV (Fig. 3, see Supplementary Fig. 9 for F1 scores, AUROC and AUPRC). Most subgroups met an equalised odds difference threshold of 0.1 (Supplementary Fig. 9). Within elective admissions, exceptions included predictions made on Sunday, patients admitted from other hospital providers, with ≥ 10 days prior length of stay and those who died in hospital. Predictions made on the day of admission had better than average performance. For emergency admissions, exceptions included patients > 80 years, those admitted to trauma and orthopaedics or from other hospital providers, with ≥ 4 days since admission and those who died in hospital.

### Sensitivity analyses by prediction time

Patients were more likely to be discharged between 10 am and 8 pm, and the observed proportion of patients discharged within the next 24 h slightly varied by prediction time chosen (Supplementary Fig. 10). Using models trained using data at 12 pm only, performance varied with different test dataset prediction times (12 am, 6 am, 12 pm, 6 pm, randomly throughout the day), with the best performance at 12 pm (AUROC = 0.871, 0.860 for elective and emergency admissions, respectively), followed by 6 am for elective admissions (AUROC = 0.861) and 6 pm for emergency admissions (AUROC = 0.812). Performance was lowest predicting discharges over the next 24 h at 6 pm for elective admissions (AUROC = 0.817) and 12 am for emergency admissions (AUROC = 0.789) (Fig. 4).

Compared to models trained and tested at 12 pm, the performance was slightly worse when the model was trained and predictions were made using times drawn randomly throughout the day, with AUROC of 0.858 and 0.849 for elective and emergency admissions, respectively (Fig. 4, Supplementary Fig. 11). However, this performance exceeded that observed for most models trained at a specific time but tested at different times.

When training and predicting at the same time of day, the performance was slightly better at 12 am and 6 am for elective admissions, with AUROC of 0.874 and 0.876, respectively (vs. 0.871 at 12 pm). However, the performance was lower at 12 am and 6 am in emergency admissions, with AUROC of 0.816 and 0.819, respectively (vs. 0.860 at 12 pm). The performance was slightly lower at 6 pm for both elective and emergency admissions compared to 12 pm (AUROC = 0.847 and 0.826) (Fig. 4, Supplementary Fig. 11).

### Training dataset size and recency impact prediction performance

Fixing the test dataset to the 12 months from 01 February 2019, we evaluated the performance of models trained with data from the preceding 1-24 months, i.e., ranging from only 01 January 2019 to 31 January 2019 to the entire period from 01 February 2017 to 31 January 2019. Individual patient-level and hospital-wide performance all improved as the number of months of training data increased, but largely plateaued after 12 months (Fig. 5a). This saturation effect suggested that the 24 months of training data used in the main analysis was more than sufficient to achieve optimal performance. The relative percentage change in AUROC for 1 vs 24 months, and 12 vs 24 months of training data was 8% and 0.9% for elective admissions, 5% and 0.6% for emergency admissions, and in AUPRC 24% and 2.1%, and 10% and 1.6% for elective and emergency admissions, respectively.

Mimicking real-world implementation, we also considered the impact of decreasing the recency of training data (Fig. 5b). We used the same fixed test dataset, but only 12 months of training data varying with an interval of 0 to 12 months between the end of the training period and the start of the testing period. Performance was generally consistent, with the most recent training data performing only slightly better. The relative percentage change for 0 vs 12-month intervals in AUROC was only 1% and 1%, and AUPRC was only 5% and 2% for elective and emergency admissions, respectively.

### Feature importance

The top five most important features for predicting discharge in elective admissions were number of oral medications received in the last 24 h, the

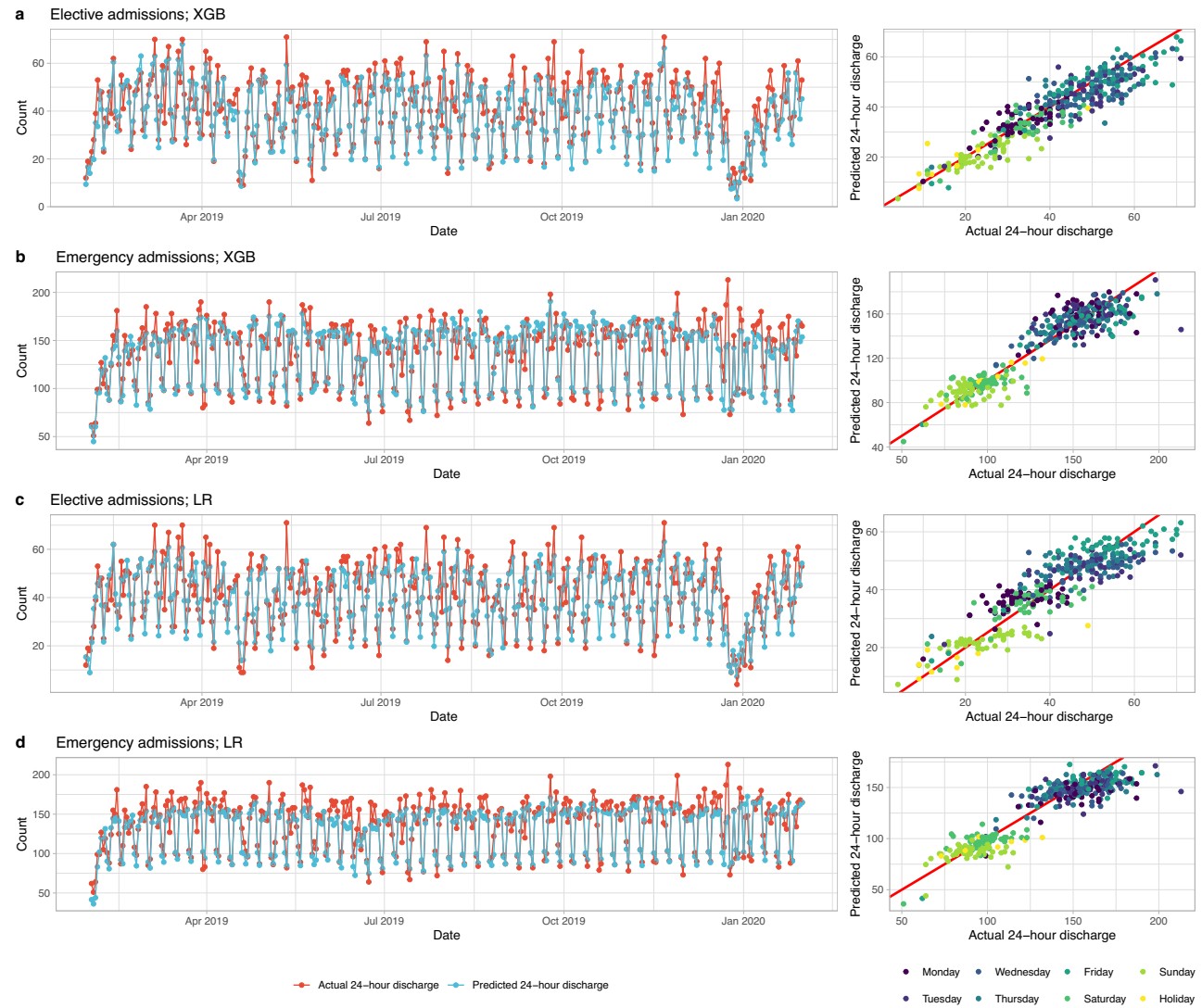

**Fig. 2 | Predicted and actual number of discharges within 24 h by calendar time in the test dataset (01 February 2019 to 31 January 2020). a** Elective admissions using extreme gradient boosting (XGB) model. **b** Elective admissions using baseline logistic regression (LR) model. **c** Emergency admissions using XGB model.

**d** Emergency admissions using baseline LR model. The distribution of errors is summarised in Supplementary Fig. 7. Predicted data from the LR models with 200 features were similar to the baseline LR models and therefore are not shown.

standard deviation of historic length of stay for other patients on the current ward, if the patient completed a course of antibiotics in the last 24 h, receipt of intravenous antibiotics in the last 24 h, and the number of procedures the patient underwent in the last 24 h. For emergency admissions, the five most important features were number of oral medications in the last 24 h, completion of antibiotics in the last 24 h, hours since admission, the standard deviation of historic length of stay for other patients on the current ward, and receipt of intravenous antibiotics in the last 24 h. The top 20 most predictive features are shown in Fig. 6 (direction of associations are shown in Supplementary Fig. 12). We also grouped individual features by feature category and summarised the mean importance of the top five most important features within each feature category. The most important feature categories were (non-antibiotic) medications, antibiotics, hospital capacity factors, procedures, and lab tests in elective admissions, and (non-antibiotic) medications, antibiotics, hospital capacity factors, demographics, and current admission factors in emergency admissions (Fig. 6). Combining elective and emergency admissions into a single model, the most important feature categories were (non-antibiotic) medications, antibiotics, hospital capacity factors, current admission factors, and ward stay features (Supplementary Fig. 13).

## Discussion

Machine learning underpinned by large-scale EHR data has the potential to transform how healthcare is delivered, but applications to the operational management of hospitals are largely unexplored[27]. By exploiting a wide range of features in EHRs, we could accurately predict patient discharge events within the next 24 h across diverse whole hospitals. We predicted the total number of discharges each day following an elective admission with an MAE of 8.9% and 4.9% following an emergency admission. We also achieved accurate predictions for individual patients with an AUROC of 0.871 and 0.858 for elective and emergency admissions, respectively. PPV and NPV were 0.555 and 0.911 following an elective admission, and 0.571 and 0.912 following an emergency admission. We achieved substantially better performance at predicting individual discharge than the baseline LR models with only a limited number of features. Both the addition of extra features and use of a more sophisticated modelling approach (XGB) improved performance, with the former having the greatest effect.

We built on prior studies and included a wide range of features, selected and curated by experienced clinicians, statisticians, and machine learning experts. We accounted for individual patient factors, both immediate and longer-term, and considered hospital-wide factors including historical

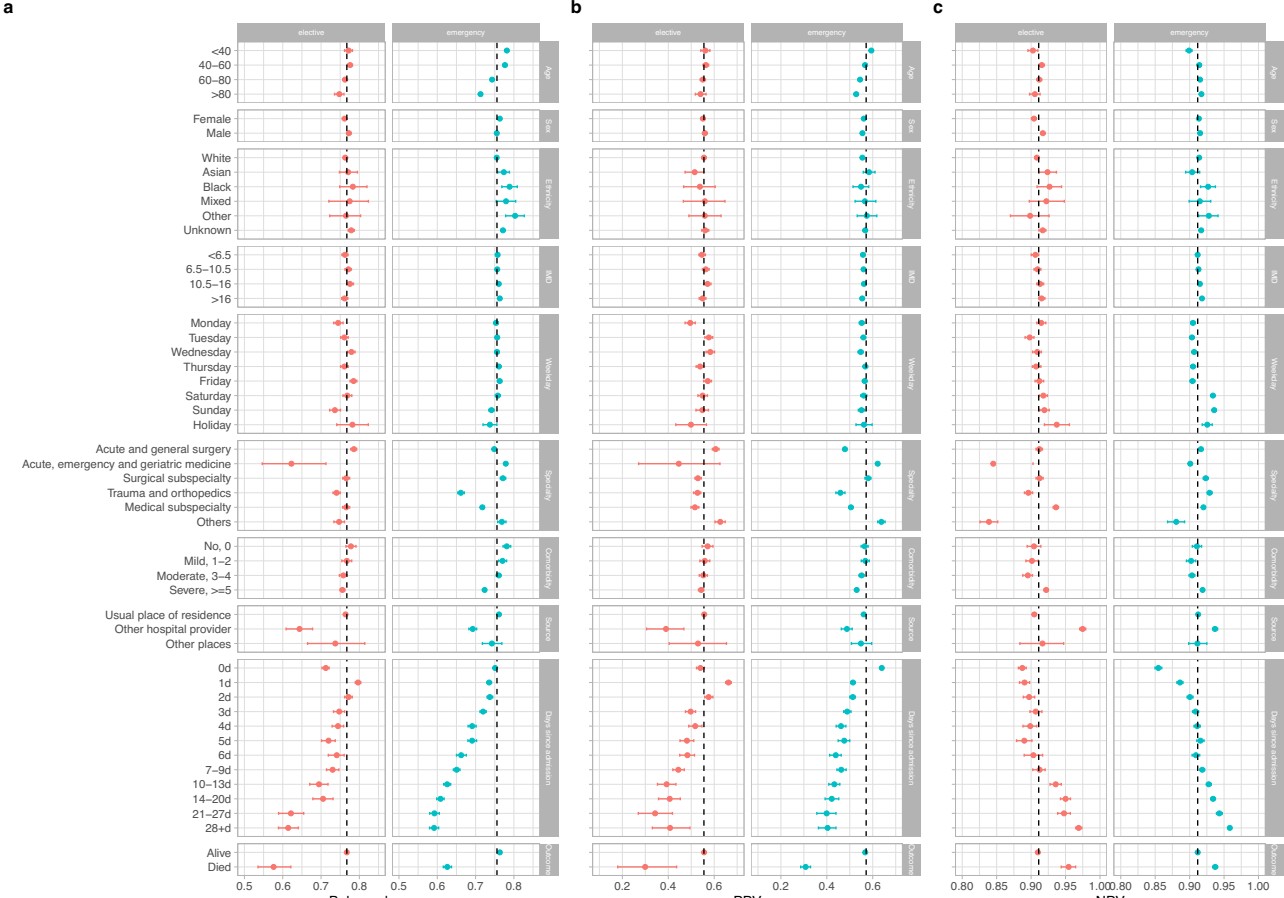

**Fig. 3 | Model performance by subgroups in the test dataset (01 February 2019 to 31 January 2020).** Balanced accuracy (**a**), positive predictive value (PPV) (**b**), and negative predictive value (NPV) (**c**) were compared. IMD=index of multiple deprivation score (higher scores indicate greater deprivation). 'Weekday' refers to the day of the week of the index date. Comorbidity was calculated using Charlson comorbidity score. 'Source' refers to the source of admission. Overall performance is shown by the dashed line in each plot. 95% confidence intervals were calculated using bootstrap (n = 500). F1 score, area under the receiver operating curve (AUROC), and area under the precision-recall curve (AUPRC) are shown in Supplementary Fig. 9.

length of stay for specific conditions. Reflecting the features used, and also the modelling approach taken, our model performance is amongst the best reported for this task, while also pragmatically accounting for patients admitted more than once and those who died, in contrast to the best-performing reported model which excluded both these patient groups[10]. Alternative model architectures, including those that allow entire time series as inputs, might further enhance performance, but may also need improved data quality to achieve significant performance gains as well as additional computational resources and training time.

We aggregated calibrated patient-level probabilities into precise predictions of daily discharge numbers at the hospital level. This approach underpins the performance achieved, and is in contrast to other approaches modelling total discharges for the entire population[28] or combining binary patient-level predictions (e.g., admission/discharge yes/no)[29]. Splitting data by calendar time rather than randomly splitting data into training and testing, we showed that hospital data in previous years could accurately predict discharges in future years. Also, our model was generally robust in predicting discharges following the start of the COVID-19 pandemic, with the performance for elective patients remaining essentially unchanged. However, performance was lower in emergency admissions, likely in part reflecting changes in reasons for admission, hospital capacity[30] and the availability of community support following discharge.

Our model performance was consistent across different population subgroups by sex, ethnicity, and deprivation, but was lower in longer admissions, older patients, and those who were admitted from other hospital providers. We achieved best performance for short-stay patients, where

factors related to their active treatment and response were important in discharge decisions and were relatively well captured by the data we used. In contrast, for longer-stay and older patients, particularly after an emergency admission, gaps in available data led to less accurate predictions. This highlights the importance of extending the data types available to improve performance, e.g., assessments of functional state which are often documented only as free text and were not available in the data we used, and external factors such as the availability of social care and typical waiting times for social care packages or residential care. Performance was much lower in patients who died in hospital, potentially because the model training favoured features that predicted recovery rather than deterioration when predicting discharge. Fitting a multiclass model, e.g., predicting discharge alive in 24 h, death before discharge, and discharge alive after 24 h, could address this, since these different outcomes are likely to have different predictors. This might substantially improve individual-level predictions whilst having less impact on overall accuracy based on summing individual-level predicted probabilities of discharge alive and death across the population.

Model performance was better with increasing training data size, with saturation at around 12 months' training data, and was slightly better using more recent than distant data, suggesting that training could be undertaken without using excessive historical data and could simply be updated two or three times a year in a real-world application. Additionally, differences between the time of day that models were trained and tested on had a relatively modest impact, however some variations exist, especially for emergency admissions with the best performance for models trained and

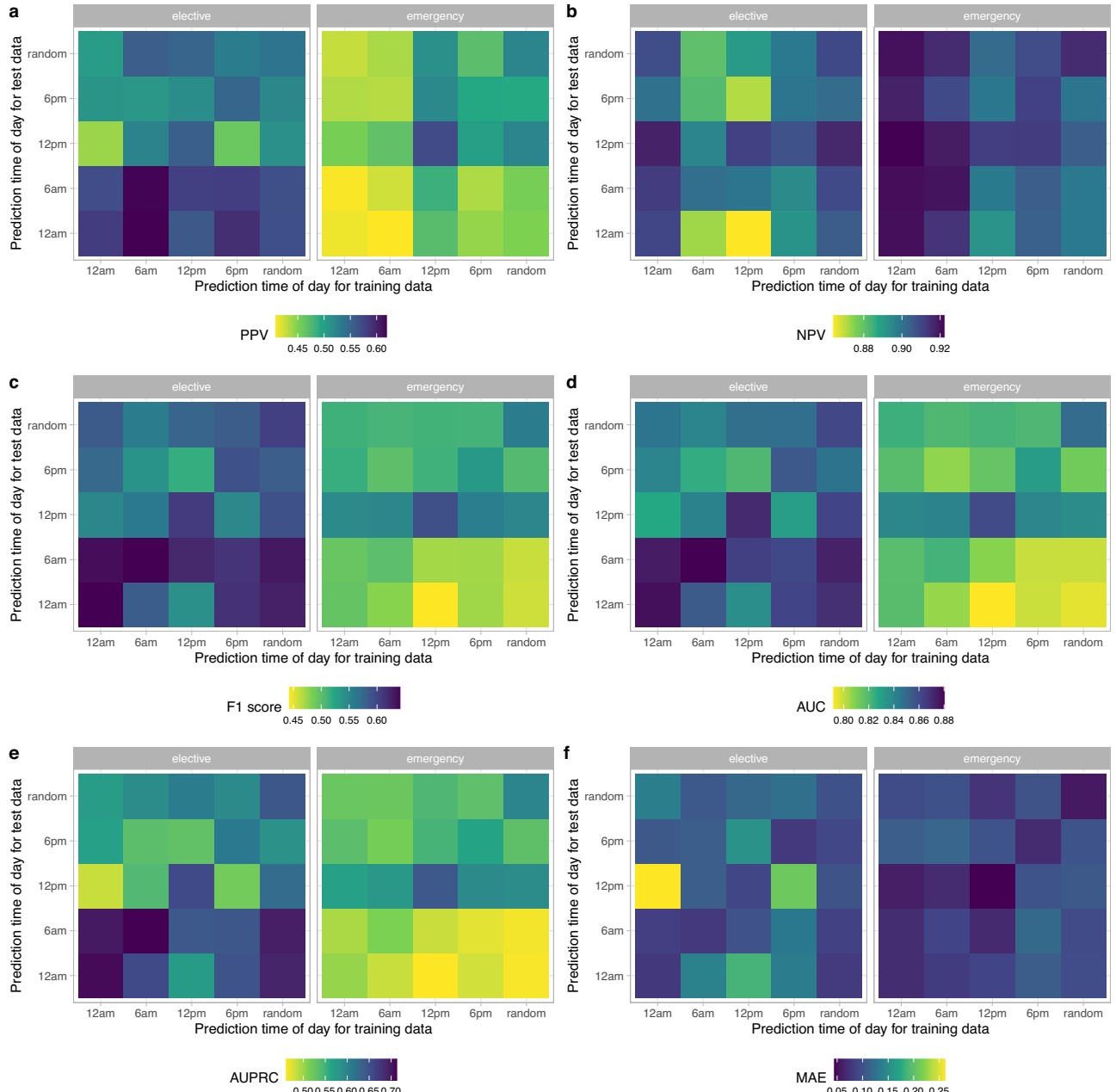

**Fig. 4 | Model performance using different prediction times of day for training and test data. a** Positive predictive value. **b** Negative predictive value. **c** F1 score. **d** Area under the receiver operating curve (AUROC). **e** Area under the precision-recall curve (AUPRC). **f** Normalised mean absolute error (MAE) (mean difference in predicted and actual discharges per day divided by the mean number of discharges per day).

tested at 12 pm. This probably reflects in part that recent data arising from clinical reviews and routine tests predominantly conducted in the morning inform these predictions. Additionally, admission of new and more unwell patients during the late afternoon and evening may make overnight and early morning predictions more challenging. Where computational resources allow, optimal performance could be achieved by using models updated throughout the day, optionally also tuned to specific times of day.

We found that non-antibiotic medications, antibiotics, and hospital capacity factors were the most predictive feature categories for both emergency and elective admissions. Switching to oral medications or completing antibiotic courses usually indicates clinical improvement, and hospital capacity factors such as the length of stay of current inpatients and the number of patients in the current ward reflect the crowdedness/pressure on hospital beds. On the contrary, discharge planning (physiotherapy contacts), microbiology tests, and previous lengths of stay and readmissions

played a relatively minor role in discharge prediction. Sequentially including more of the most predictive features increased model performance, but plateaued after including the top 200 features, and the computational time was substantially reduced to only 20% vs when all 1152 features were used, suggesting that including more features does not necessarily improve performance. Hospitals may need to pay more attention to the data collection quality of the key features that are most predictive to achieve accurate and efficient predictions.

Ensuring the availability of beds and timely patient discharge is pivotal in managing patient flow within healthcare systems[31], but existing interventions often adopt static procedures such as 'discharge by noon', or respond only to critical levels of patient demand[32], therefore failing to address the broader complexities of patient flow dynamics. Moreover, the integration of automated discharge prediction models into clinical practice remains limited[33]. Although clinician predictions of discharge within 24 h

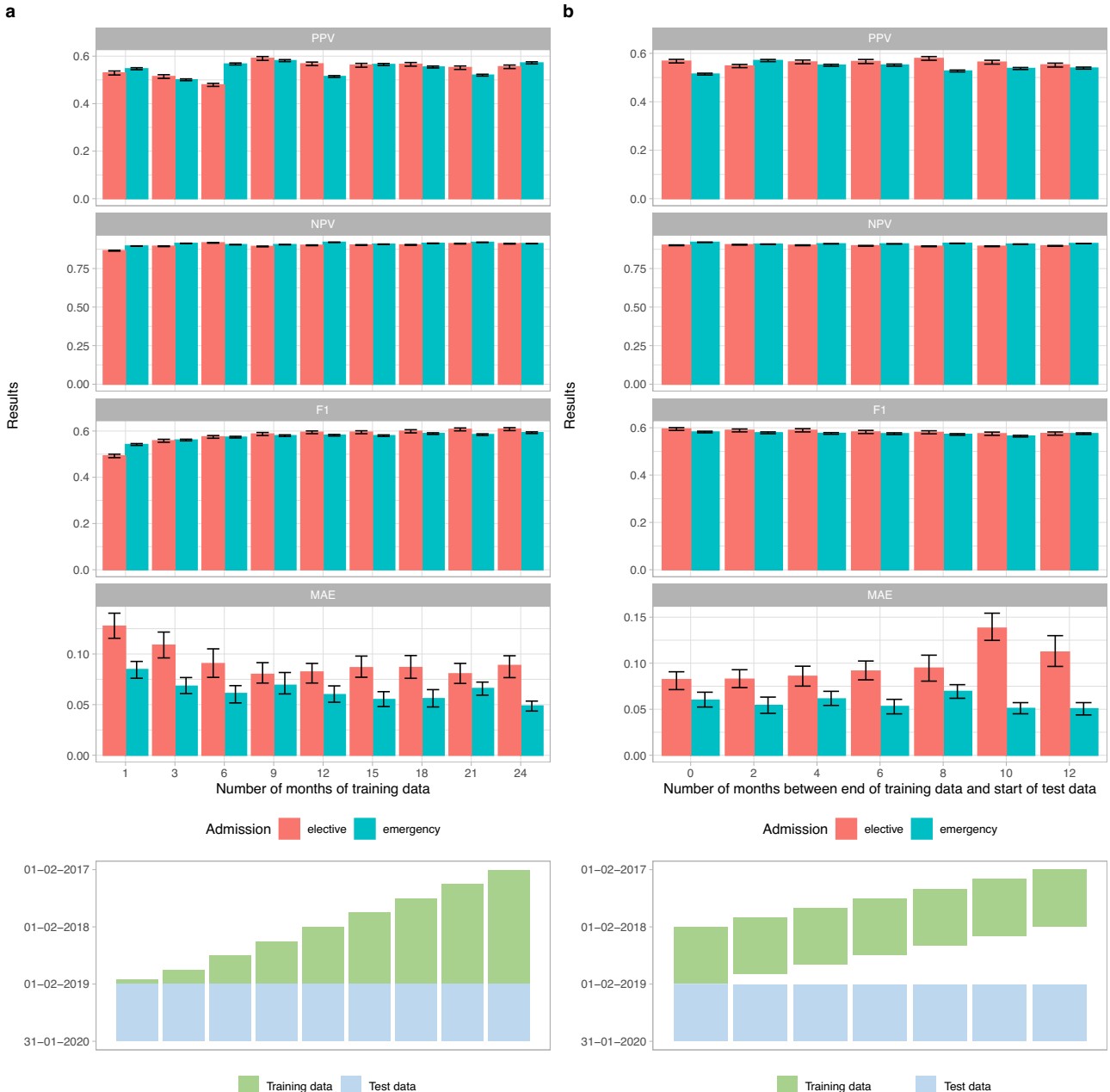

**Fig. 5 | Impact of training dataset size and recency on discharge prediction performance in the test dataset (01 February 2019 to 31 January 2020). a** By increasing training data size. **b** By decreasing recency of same-size training data. 95% confidence intervals were calculated using bootstrapping (*n* = 500). PPV: positive predictive value, NPV: negative predictive value, MAE: normalised mean absolute error (mean difference in predicted and actual discharges per day divided by the mean number of discharges per day).

were not available in our data, previous studies have shown that prediction models performed as well or more accurately than clinicians in predicting hospital discharges, demonstrating the potential for operational benefits[11,12]. A few previous studies have also implemented their discharge prediction models and observed a reduction in hospital length of stay[12,34], we therefore envision our approach could be deployed in several ways. For example, within each hospital ward, a dashboard of discharge tasks (confirming follow up plans, writing discharge letters, preparing discharge medication, booking transport home, social care readiness, room cleaning, etc) and binary predictions of discharge within the next 24 h, could be used by healthcare workers to flag patients likely to be discharged to facilitate timely completion of outstanding tasks. Discharge probabilities could also be used to rank the patients most likely to go home to ensure their discharge preparations were prioritised. Where the total number of planned or expected

admissions within the next 24 h exceeded the predicted number of discharges, preparations for redeploying staff and resources could be made by operational teams. However, further trials using the best performing models and different implementation strategies are required to establish optimal approaches, alongside economic evaluations to address what level of model accuracy would be required (e.g. avoiding having to cancel booked transport or cleaning) for these interventions to be cost-effective and resource efficient. On-going oversight of input data quality/completeness and the accuracy of predictions by specialist data science teams would be required at a hospital level. Taken together, our models have the potential to be applied to improve the efficiency of patient flow across hospital settings, leading to accelerated care delivery and patient recovery, and optimal use of healthcare resources. Although our study was based on data from hospitals in Oxfordshire, this framework could potentially also apply to hospitals in

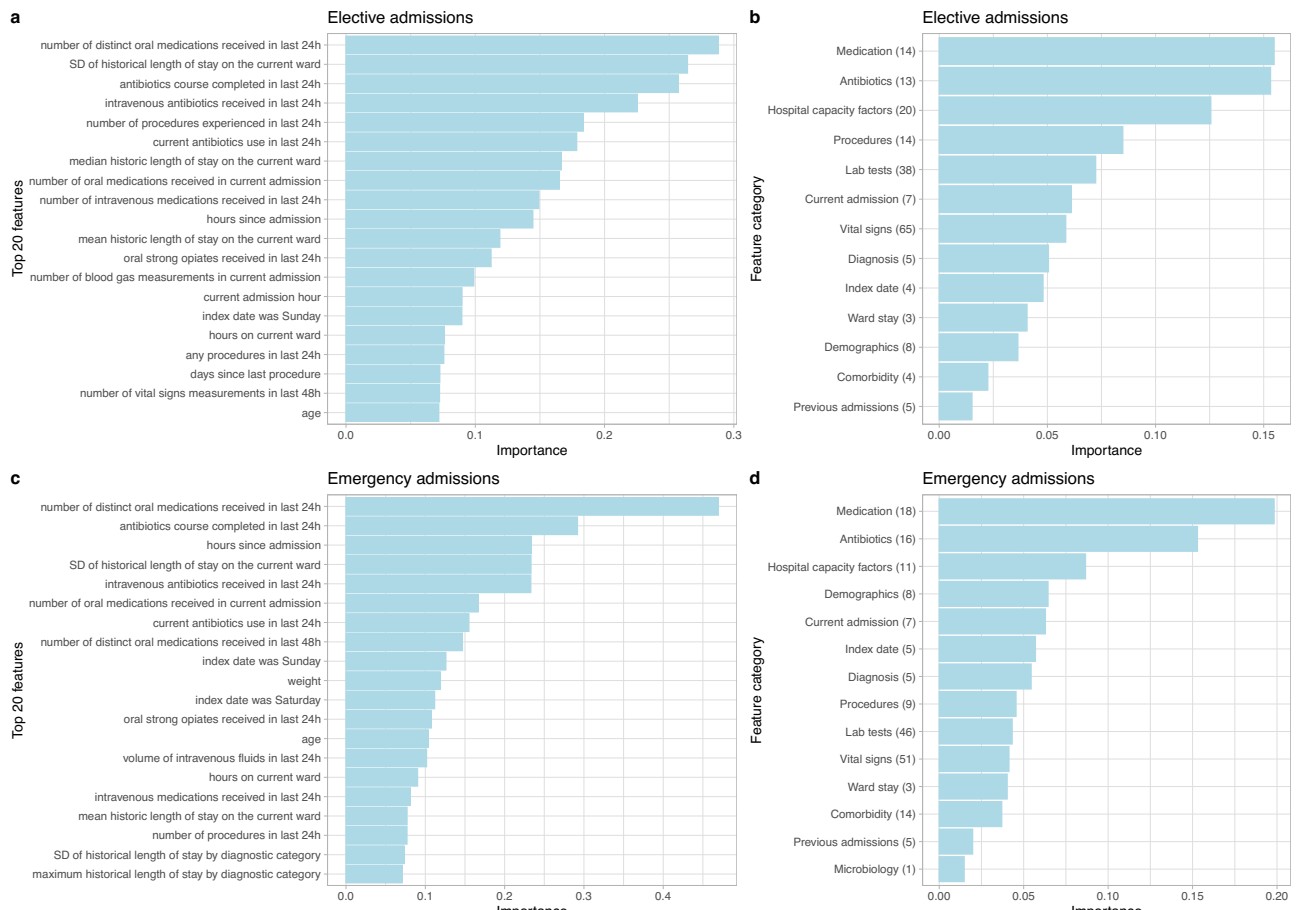

**Fig. 6 |** Feature importance from extreme gradient boosting models using SHAP values for elective admissions (**a**, **b**) and emergency admissions (**c**, **d**). The top 20 most predictive features are shown in the order of predictiveness in (**a**, **c**). The direction of association between the top 20 features and discharge are shown in Supplementary Fig. 12. Feature importance grouped by feature category is shown in the order of predictiveness in (**b**, **d**). The mean importance of the top 5 most important features within each category is plotted. Numbers shown in parenthesis are number of features within the top 200 most predictive features in each category. No microbiology features were selected for elective admissions, and no discharge planning features were selected for both admission types. The complete list of features is summarised in Supplementary Table 1. SHAP: SHapley Additive exPlanations, SD: standard deviation, Current admission: admission time/source/specialty, Diagnosis: length of stay characteristics of diagnostic categories; Previous admissions: previous length of stay and readmission; Discharge: discharge planning.

other regions within the UK and internationally with similar settings, offering a prospect for mitigating the pressure of overcrowding on NHS and other healthcare systems, thus improving healthcare delivery efficacy and resource management on a national scale.

Limitations of our study include our focus on a relatively short prediction horizon of 24 h, limiting the scope of planning and interventions that could follow the same timescale. However, our approach could be adapted to make longer-range predictions too. We focused on ordinary admissions, excluding day cases and other regular admissions to avoid artificially inflating performance as the expected length of stay was known in advance for these patients. However, some of these admissions may unexpectedly have stayed longer than expected and this is not captured by our current approach. A strength of our approach is that it produced a generalised model for the whole hospital, however it is possible that refined models for specific patient subgroups, e.g. based on prior length of stay, procedure or diagnosis, could offer improved performance albeit leading to greater complexity in model training, oversight and implementation. To some extent the XGB architecture addresses this within a single model, as key patient groups can be separated by the decision tree fitting process if this improves performance, with each patient subgroup having distinct downstream decision trees.

We used diagnosis categories derived from ICD-10 codes for training and testing, which were only recorded at discharge, however in reality the primary working diagnosis is known in real-time to clinicians and could be used if documented electronically. We did not incorporate hospital features such as the percentage of occupied beds, as the number of available beds in each hospital was not available in our dataset. Additionally, we did not evaluate how model performance varied with the degree of operational pressure experienced by hospitals, which could be considered in the future as accurate predictions are particularly beneficial when resources are most constrained. We adopted a relatively simple, pragmatic approach to feature engineering, using summaries of time series data including blood tests and vital sign measurements. Performance could potentially be improved by better representing these dynamic data in future work, and by investigating the benefits of dimensionality reduction and more complex feature representations such as *t*-SNE[35] or autoencoders[36]. Moreover, we used temporal data from the same hospital for validation and testing, rather than data from a completely different hospital group. Future work should incorporate validation with data from diverse settings to further strengthen the validity and generalisability of our findings, as well as studies of the impact of deploying similar models.

We only used structured EHR data for predicting imminent hospital discharge and did not consider other data types such as unstructured free text, which could potentially improve prediction further, particularly for patients with prolonged hospital stays. We used the XGBoost algorithm, which is a widely employed method recognised for its superior performance compared to other traditional machine learning models, but other advanced architectures including deep learning models have the potential for

improving performance and accommodating flexible updates, such as incorporating new data over time or across different settings. In recent years, there has been an increasing interest in using natural language processing and deep learning models for hospital management tasks such as length of stay prediction and utilising more complex data including free text medical records[37–39]. For example, a recent study demonstrated the efficacy of large language models trained on unstructured clinical notes for predicting hospital length of stay, outperforming traditional models[40]. Future studies should explore similar approaches and make use of unstructured data to enhance predictive capabilities for healthcare management.

In conclusion, our study shows the feasibility of integrating machine learning modelling approaches with EHR data to facilitate real-time operational management in hospitals, with realistic requirements for training data and model updating. Our models achieve a good performance for both individual-level and hospital-level discharge predictions, demonstrating the potential to be deployed to improve the efficiency of hospital management, patient flow dynamics, and expedite patients' recovery and discharge processes.

## Data availability

The datasets analysed during the current study are not publicly available as they contain personal data but are available from the Infections in Oxfordshire Research Database (https://oxfordbrc.nihr.ac.uk/research-themes/modernising-medical-microbiology-and-big-infection-diagnostics/iord-about/), subject to an application and research proposal meeting the ethical and governance requirements of the Database. For further details on how to apply for access to the data and a research proposal template please email iord@ndm.ox.ac.uk. The source data for Figs. 1–6 is located in Supplementary Data 1–6.

## Code availability

A copy of the analysis code is available at https://github.com/jiaweioxford/discharge_prediction. https://doi.org/10.5281/zenodo.14015332[41].

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

## Acknowledgements
This study was funded by the National Institute for Health Research (NIHR) Health Protection Research Unit in Healthcare Associated Infections and Antimicrobial Resistance at Oxford University in partnership with the UK Health Security Agency (UKHSA) and the NIHR Biomedical Research Centre, Oxford. The views expressed in this publication are those of the authors and not necessarily those of the NHS, the National Institute for Health Research, the Department of Health and Social Care or the UKHSA. DAC was supported by the Pandemic Sciences Institute at the University of Oxford; the National Institute for Health Research (NIHR) Oxford Biomedical Research Centre (BRC); an NIHR Research Professorship; a Royal Academy of Engineering Research Chair; the Wellcome Trust funded VITAL project (grant 204904/Z/16/Z); the EPSRC (grant EP/W031744/1); and the InnoHK Hong Kong Centre for Cerebro-cardiovascular Engineering (COCHE). DWE is a Robertson Foundation Fellow. The funders had no role in in study design; in the collection, analysis, and interpretation of data; in the writing of the report; or in the decision to submit the paper for publication. This work uses data provided by patients and collected by the UK's National Health Service as part of their care and support. We thank all the people of Oxfordshire who contribute to the Infections in Oxfordshire Research Database. Research

Database Team: L Butcher, H Boseley, C Crichton, DW Crook, DW Eyre, O Freeman, J Gearing (public representative), R Harrington, K Jeffery, M Landray, A Pal, TEA Peto, TP Quan, J Robinson (public representative), J Sellors, B Shine, AS Walker, D Waller. Patient and Public Panel: M Ahmed, G Blower, J Hopkins, V Lekkos, R Mandunya, S Markham, B Nichols. Members of the panel contribute to setting research priorities and questions, and have reviewed a summary of the data used and of the analytical approach. All results are shared with the panel for comment. We would also like to thank Lisa Glynn and colleagues at Oxfordshire University Hospitals NHS Trust for helpful feedback and insights into early versions of the data analyses presented.

## Author contributions
The study was designed and conceived by D.W.E., A.S.W., D.A.C., and A.J.B.; J.W., D.W.E., and J.Z. curated the data. J.W., J.Z., Z.Z., K.Y., Q.G., and A.L. analysed the data and created the visualisations. J.W., J.Z., and D.W.E. wrote the first draft of the manuscript. All authors contributed to editing and revising the manuscript.

## Competing interests
D.A.C. reports personal fees from Oxford University Innovation, outside the submitted work. No other author has a conflict of interest to declare.
