## [Transparent Peer Review file · Communications Medicine]

Predicting individual patient and hospital-level discharge using machine learning

Corresponding Author: Professor David Eyre

Version 0:

Reviewer comments:

Reviewer #1

(Remarks to the Author)
Dr Jonathan M Clarke
Imperial College London

Thank you for giving me the opportunity to review this well written and well conducted study applying extreme gradient boosting (XGB) models to predict discharge from acute hospitals within the next 24 hours.

Though I have significant experience in the use of EHR datasets in health services research, I would not consider myself an expert in XGB models and recommend review from an expert in this and related methods.

The authors have done an excellent job of conveying a range of technical concepts in a very clear and structured way, making what could have been a tough paper to read enjoyable. I am also very impressed by their efforts to incorporate clinical expertise in the selection and cleaning of input data, and their commitment to the eventual clinical utility of the work.

While the study appears to have been well executed, there are some areas I feel the authors could consider further to improve on the clarity and interpretation of the work.

Firstly, the inclusion of hospital-level discharge accuracy is often overlooked and crucially important to the operations of acute hospitals. While the authors quantify the average error of daily hospital discharge prediction, it may be useful to understand the distribution of this error – e.g. how often are estimated far off the mark, are they consistently above or below expected. These may help a reader better understand how usefully such a model may be applied in practice. The scatter plots in Figure 2 show this, but it is not really expanded on.

Using 24-hour discharge as an outcome pitches the model against the expertise of ward nurses and hospital discharge teams who are often asked to make such predictions, and may have access to greater information on which to predict likely discharge than structured data in the EHR. While the authors mention the role of clinicians in determining discharge, perhaps greater discussion of this may be useful. E.g. Barnes et al. (2016) compares the performance of clinicians to LR and RF models.

The choice of a logistic regression model as a comparison for a simple model is useful, however I do wonder whether you are perhaps flattering the performance of the XGB model through including far fewer parameters in the logistic model, the majority of which are not the most informative in the XGB. In a sense, the study is comparing the two models to simultaneously examine the benefit of including far more data into a predictive model, and a more flexible / sophisticated technique. Perhaps the authors could consider either a simpler XGB model to match the LR, or increasing the parameters fed into an LR as comparison? Disentangling 'more data' from a 'better technique' would be important for those considering using the findings of the study. From looking at the scatter plots in Figure 2, neither model does a great job at predicting overall hospital discharges, with the differences between LR and XGB quite hard to see.

Line by line comments:

Line 62: Prompting clinicians that a patient is approaching readiness for discharge based on a model derived from structured EHR data is a challenging one suggestion, and should be made cautiously. What sort of level of model accuracy would be required for each of these use statements, if enacted, to be efficient rather than inefficient (e.g. cleaners readied inappropriately, transport being cancelled).

Line 93: Consider clarifying what is meant by 'health inequalities' in this context.

Line 112: The authors mention the data are sourced from four teaching hospitals. Are the hospital capacity measures (e.g. number of inpatients) calculated for each hospital independently, or across the system as a whole? Further, does the hospital a patient is admitted to influence their length of stay? How common are interhospital transfers between the trusts? If total discharges are determined across the system, such patients would count as discharged while remaining an inpatient at another of these four hospitals.

Line 113: How were these case definitions made? Sometimes day cases stay over, or a regular admission has an unplanned extension to their stay.

Line 140: Were the laboratory results included according to their numerical results or their binary normal or abnormal state? As the study deals with decisions to discharge, normality or abnormality, rather than numerical value, may be a better predictor, as that, rather than the number per se, is likely to be what clinically influences a discharge decision.

Line 153: Were arterial, venous and capillary blood gases treated separately or together?

Line 159: How was this preprocessing done? Based on clinical expertise? How much data was removed here?

Line 163: Consider expanding on the assertion that XGB can handle missing data by default. It strikes me as surprising that if e.g. age was missing 99% of the time but is expected to influence length of stay it can be handled well by any model. I think some understanding of how often key variables were missing is important regardless of the capacity of the method to handle missingness.

Line 274: Including a frequency plot of length of stay for emergency and elective patients separately may be useful for the reader to understand the underlying data you are predicting based on.

Line 363: This is an interesting finding, and one that has some plausibility based on patient flow through hospitals. Generally elective patients arrive for around 7am for planned operating lists, while emergency patients may arrive at any time of day. Patients admitted in a planned fashion e.g. the night before an operation are usually clinically more complex, and thus may have harder to predict lengths of stay. Anecdotally (from my practice at least), acuity of emergency presentations varies over the course of the day, with higher acuity dominating at night (i.e. things that couldn't wait til morning). These features can be seen, particularly for elective care, in Figure 4. Perhaps consider providing explanations for the differences you see across your time comparisons, given you include them and find some variation.

Line 407: When discussing feature importance, it may be helpful for the reader to understand the direction of any association between the variable(s) and the outcome – e.g. does receipt of IV antibiotics in the last 24 hours increase or decrease likelihood of discharge? I also see here that the SD of historic length of stay is important. Given the likely skew of LOS in the data, could quantile measures, such as using 25th and 75th centiles be helpful here and avoid the enforced symmetry of the SD?

Line 412: How were the procedures included defined? How was the number of procedures quantified (e.g. if using OPCS codes for laparoscopic surgery may have two codes – one to describe the operation and another the approach)? Were minor and major procedures differentiated? E.g. a simple procedure has less of an impact on LOS than a major one.

Line 494: Different variables are likely to have different importance for different patient groups and admissions. As you are pooling a wide range of patients and types of admission, is there a chance that the dominance of common admissions reduces the predictive accuracy for rarer types of presentation? While performance plateaued after 200 features, as there further performance gains for some patient groups for including more features? It seems from the results like a major challenge is predicting when someone who has been in hospital for a long period is likely to go home. I imagine features predictive of their discharge are different to the discharge of someone who has been in hospital for 1 day.

Line 616: what does 'community' mean here?

Line 732: Consider expanding acronyms and abbreviations, particularly if not done in the main text already.

Line 776: These plots are great, and useful to see. In most cases, a clear difference between weekday and weekend predictions is seen. The predicted values appear markedly flatter than the actual values observed for the LR, sometimes quite markedly, but perform slightly better for the XGB. In both cases, comparing the LR and XGB plots side by side shows neither are necessarily excellent predictors of discharge numbers.

(Remarks to the Author)

SUMMARY

The paper develops and validates a model for patient discharge prediction in a large UK hospital network. Overall, the analysis is thoroughly conducted, exhaustive, and clearly explained. The authors also provide a range of sensitivity analyses (using prediction times on the test set different than those used for training; analysing the impact of the size and recency of the training data on out-of-sample performance; comparing pre- and post-COVID performance) that are insightful and, as far as I can tell, not common in the literature.

I would support the publication of the manuscript upon minor revisions.

My primary concerns are:

- Misleading title. The present study does not demonstrate/support any improvement on patient flow. This is pure speculation (based on evidence from other studies). The present paper is about the development and the validation of a machine learning model for discharge prediction. It should be advertised as such.
- Simplistic baseline. The logistic regression baseline is overly simplistic in my opinion. A more credible benchmark should be used (see detailed comments)

DETAILED COMMENTS

Title "Improving patient flow through hospitals with machine learning based discharge prediction" Nothing in the study justifies the claim that patient flow "improved". This study is a retrospective analysis, evaluating the predictive power of ML models for discharge prediction. The fact that these predictions could improve patient flow is not supported in this study.

Please edit the title to reflect the contribution of the present paper more faithfully

p.2 Abstract l.36 "AUCs of 0.87" -> "AUROCs of 0.87" (since the authors use 2 types of AUC metrics)

p.2 Abstract l.44 "optimising" Please check journal policy on the use of UK or US English.

p.3 l.62 "e.g. prompting" -> e.g., prompting (general comment)

p.4 l.94 "Most studies either evaluated individual-level discharge prediction performance or hospital-wide predictions, but did not combine the two in a single approach." The term "most studies" suggests that some studies did. It would be good to have a more faithful and precise description of the literature here. Such as "As far as we know, only Study XX did ..." or "With the exception of X, Y, all of the aforementioned studies..."

p.4 l.112-118 Exclusion criteria. Why did the authors need to exclude both patients <= 16 yo and paediatric patients. I would think of those criteria as redundant. Are they? If not, why?

p.5 l.163 "because extreme gradient boosting (XGB) models can handle missing values by default" The authors should specify which package/implementation of XGB they used. Different software packages implement different strategies for dealing with missing values. See Section 5 in Josse, J., Prost, N., Scornet, E., & Varoquaux, G. (2019). On the consistency of supervised learning with missing values. arXiv preprint arXiv:1902.06931.

p.5 l.164 "For the baseline logistic regression model, the included features (age, sex, day of the week, and hours since admission) did not have missing values." This is one of my biggest concerns about the study. The baseline is overly simplistic. I recommend the authors use (a) a regularised (L1 or L2) logistic regression model; (b) a simple imputation method (mean imputation for continuous, new category for categorical features)—note that this is not more complicated than the "average LOS for the diagnosis code" feature they are using; (c) the same set of features and overall training pipeline as their XGB model (Figure 1.d). This would be a fair comparison.

p.7 l.242 "are likely to apply to hospitals with similar daily discharge rates" What is the actual discharge rate in the hospital studied? It would be important to share these numbers earlier for the reader to appreciate the representativeness of the hospital

p.9 l.327 "We calculated the total number of discharges expected in the next 24 hours across all elective or emergency admissions in the hospital by summing the individual-level predicted discharge probabilities." This explanation should be provided earlier, in the methods section (e.g., around p.7 l.247). In particular, with this aggregation strategy, it is important that the authors have calibrated their predicted probabilities and not use their binary predictions with the F1-optimising threshold (p.7 l.236). So, the authors should highlight these points.

p.9 l.363 "Sensitivity analyses by prediction time" I am a little bit confused with this sensitivity analysis. The models have been trained to predict whether a patient, based on information on day d at noon, will be discharged on day (d+1) at noon (within 24 hours). Am I correct that the different prediction times (midnight, 6am, 6pm) correspond to the following prediction horizon: 12 hours, 18 hours, 30 hours? It would be good if this could be better explained in the paper. Also, does it make sense that the model performances are more robust to some prediction time than others?

p.9 l.364 "Patients were more likely to be discharged between 10 am and 8 pm," I am surprised that this observation only comes now in the study. For me, it should have informed the design of the study from the beginning. This pattern is very common and it is thus quite odd to use noon as the baseline prediction time in my opinion. Also, from an implementation perspective, noon is a very busy time in the hospital so it might be hard to properly extract data from the EHR in real-time to generate the features used for the prediction (this is one of the main reasons why deployed ML models are typically run overnight, when EHR activity is lower).

p.9 l.364 "Patients were more likely to be discharged between 10 am and 8 pm," Is there some heterogeneity in that observation between elective and emergent admissions? (That could explain heterogeneity observed in line 380-385)

p.10 l.407 "Feature importance" I am surprised not to see current length of stay as an important predictive feature for elective admission. Was it included in the study? Was it selected at least as part of the top 200 features? Given the importance of that feature in other studies from the literature, the authors should comment.

p.11 l.452-453 "further studies comparing model performance to clinician predictions and of trial implementations are required." The authors do not give enough credit here to studies that have done that, e.g., [7,13] or King, Z., Farrington, J., Utley, M., Kung, E., Elkhodair, S., Harris, S., ... & Crowe, S. (2022). Machine learning for real-time aggregated prediction of hospital admission for emergency patients. NPJ Digital Medicine, 5(1), 104.

Na, L., Carballo, K. V., Pauphilet, J., Haddad-Sisakht, A., Kombert, D., Boisjoli-Langlois, M., ... & Bertsimas, D. (2023). Patient outcome predictions improve operations at a large hospital network. arXiv preprint arXiv:2305.15629.

p.12 l.513 "We envision our approach could be deployed in several ways". Again, this vision is largely based on how previous studies have deployed their algorithms. Credit should be given to such studies.

Version 1:

Reviewer comments:

Reviewer #1

(Remarks to the Author)

Thank you for giving me the opportunity to review a revised version of this manuscript. The authors have gone to great effort to address the comments raised by the previous version and I'm satisfied with the response in the vast majority of cases. It's very useful to see the improvement in performance in the LR models through using the same set of variables as used in the XGB model. This improvement seems to be greater than the improvement resulting from using XGB over LR (though this is not explicitly calculated). My main comment would be to ensure this finding is mentioned prominently across the manuscript, particularly in line 30 and lines 478-481. In both of these cases, the emphasis is on the difference between the baseline LR and the complete XGB models. Given the substantial improvement achieved in the LR model through simply feeding it the same data as the XGB, it think there's a danger the reader attributes the cause of the difference to the modelling method rather than the breadth of input data.

Lines 309-310: Thank you for including the supplementary figure for the distribution of LoS. I still do feel that only including the median and IQR in the main text doesn't adequately express the profound skewness of the distribution. I would recommend commending on this in the main text and linking to the figure in that context accordingly.

I congratulate the authors on a very well-conducted study that will be of interest and use to a broad readership.

Reviewer #2

(Remarks to the Author)

All of my comments have been satisfactorily addressed by the authors.

Reviewers' comments:

Reviewer #1 (Remarks to the Author):

Dr Jonathan M Clarke
Imperial College London

Thank you for giving me the opportunity to review this well written and well conducted study applying extreme gradient boosting (XGB) models to predict discharge from acute hospitals within the next 24 hours.

Though I have significant experience in the use of EHR datasets in health services research, I would not consider myself an expert in XGB models and recommend review from an expert in this and related methods.

The authors have done an excellent job of conveying a range of technical concepts in a very clear and structured way, making what could have been a tough paper to read enjoyable. I am also very impressed by their efforts to incorporate clinical expertise in the selection and cleaning of input data, and their commitment to the eventual clinical utility of the work.

While the study appears to have been well executed, there are some areas I feel the authors could consider further to improve on the clarity and interpretation of the work.

Firstly, the inclusion of hospital-level discharge accuracy is often overlooked and crucially important to the operations of acute hospitals. While the authors quantify the average error of daily hospital discharge prediction, it may be useful to understand the distribution of this error – e.g. how often are estimated far off the mark, are they consistently above or below expected. These may help a reader better understand how usefully such a model may be applied in practice. The scatter plots in Figure 2 show this, but it is not really expanded on.

Response: We have added the distribution of errors for the four models (elective-XGBoost, elective-LR, emergency-XGB, emergency-LR) as Supplementary Figure 7. We plotted the absolute error (differences in predicted and actual discharges each day) in panel a and the normalised absolute error (differences in predicted and actual discharges each day divided by the actual discharges) in panel b. We have also added further results text to highlight the typical absolute errors seen in the context of the number of emergency and elective patients in hospital and discharged each day (line 368-370).

Using 24-hour discharge as an outcome pitches the model against the expertise of ward nurses and hospital discharge teams who are often asked to make such predictions, and may have access to greater information on which to predict likely discharge than structured data in the EHR. While the authors mention the role of clinicians in determining discharge, perhaps greater discussion of this may be useful. E.g. Barnes et al. (2016) compares the performance of clinicians to LR and RF

models.

Response: We do not have clinician predictions of discharge within the next 24 hours available in our data, which we have now added as a limitation in the discussion. We have expanded our discussion and included citations for studies that compared the model predictions with clinician predictions (line 562-565).

The choice of a logistic regression model as a comparison for a simple model is useful, however I do wonder whether you are perhaps flattering the performance of the XGB model through including far fewer parameters in the logistic model, the majority of which are not the most informative in the XGB. In a sense, the study is comparing the two models to simultaneously examine the benefit of including far more data into a predictive model, and a more flexible / sophisticated technique. Perhaps the authors could consider either a simpler XGB model to match the LR, or increasing the parameters fed into an LR as comparison? Disentangling 'more data' from a 'better technique' would be important for those considering using the findings of the study. From looking at the scatter plots in Figure 2, neither model does a great job at predicting overall hospital discharges, with the differences between LR and XGB quite hard to see.

Response: We wanted to compare an easy-to-fit LR model with minimal features vs. an XGB model with complex features, as we wanted to see the additional improvement from a very simple baseline that would be accessible to most hospitals. However, we agree that we could have better disentangled having more data/features from a 'better' model architecture. Therefore, we have added the performance of an LR model with 200 features (same as the final XGB model) to the previous two models, and compared the performance across three groups. The performance of the 200 feature LR model was greater than the simple LR model, but lower than the XGB model. The difference was largest between the two LR models, indicating that having a greater number of well-designed features makes more difference than using a more flexible/sophisticated model. We have updated Table 3, and added this finding to the results (line 343-349), coming back to it again in the discussion.

Line by line comments:

Line 62: Prompting clinicians that a patient is approaching readiness for discharge based on a model derived from structured EHR data is a challenging one suggestion, and should be made cautiously. What sort of level of model accuracy would be required for each of these use statements, if enacted, to be efficient rather than inefficient (e.g. cleaners readied inappropriately, transport being cancelled).

Response: We agree this is important. However, determining necessary model accuracy for each of these use cases requires further economic modelling that is beyond the scope of the current study. We have added to the discussion the

need for further trials and cost-effectiveness evaluations as areas of future work in the paragraph where we discuss potential interventions (line 576-580).

Line 93: Consider clarifying what is meant by 'health inequalities' in this context.

Response: We have clarified this as 'different sociodemographic groups'.

Line 112: The authors mention the data are sourced from four teaching hospitals. Are the hospital capacity measures (e.g. number of inpatients) calculated for each hospital independently, or across the system as a whole? Further, does the hospital a patient is admitted to influence their length of stay? How common are interhospital transfers between the trusts? If total discharges are determined across the system, such patients would count as discharged while remaining an inpatient at another of these four hospitals.

Response: We have clarified in the methods that we summarised performance across all 4 hospitals in the OUH hospital group. It is possible that the hospital or ward that patients are admitted to influence their length of stay and we included historic length of stay on the patient's current ward as a feature in the model.

We have clarified that transfers were only those that were outside of OUH, which was 2% of discharges in total.

Line 113: How were these case definitions made? Sometimes day cases stay over, or a regular admission has an unplanned extension to their stay.

Response: Case definitions were based on how various admission codes are routinely used at our institution. We have clarified this in the methods text (line 116). We excluded day cases and regular admissions as the expected/actual length of stay was known *a priori* for nearly all and we did not want to artificially inflate performance. However, we agree that this may not be accurate all the time, e.g. day cases may unexpectedly stay overnight. We have added this as a limitation (line 594-597).

Line 140: Were the laboratory results included according to their numerical results or their binary normal or abnormal state? As the study deals with decisions to discharge, normality or abnormality, rather than numerical value, may be a better predictor, as that, rather than the number per se, is likely to be what clinically influences a discharge decision.

Response: We used the numerical values for all the laboratory features. The XGB model is an implementation of gradient boosted decision trees. A key feature of XGB is its ability to automatically identify and set cut-off points by splitting the values of input features. So this process can split the data at various points to find the optimal threshold that separates normal and

abnormal values. We have added a sentence to clarify this in the methods (line 206-208).

Line 153: Were arterial, venous and capillary blood gases treated separately or together?

Response: Arterial, venous, and capillary are combined because the recording of blood gases as arterial, venous, or capillary was not always reliable. We have clarified this in the methods (line 160-161).

Line 159: How was this preprocessing done? Based on clinical expertise? How much data was removed here?

Response: The preprocessing was based on clinical expertise. We have included a table summarising the implausible extreme limits for each vitals and lab test features as Supplementary Table 2. Values that exceeded the limits were removed during the preprocessing steps, and represented only a very small minority of the data as a whole.

Line 163: Consider expanding on the assertion that XGB can handle missing data by default. It strikes me as surprising that if e.g. age was missing 99% of the time but is expected to influence length of stay it can be handled well by any model. I think some understanding of how often key variables were missing is important regardless of the capacity of the method to handle missingness.

Response: The proportion of missingness for the 200 variables ranged from 0% to 88%. We have plotted the proportion of missingness for elective and emergency models in Supplementary Figure 2. XGB models handle missingness by 'Missing Incorporated in Attribute' (MIA), which is a method that naturally handles missing values in decision trees by using missingness itself as a splitting criterion. We have added the method for handling missing data and the proportion of missingness to the Methods (line 173-176).

Line 274: Including a frequency plot of length of stay for emergency and elective patients separately may be useful for the reader to understand the underlying data you are predicting based on.

Response: The median (IQR) of length of stay was previously summarised in Table 2 and provided in the results text. We have added the frequency plot of length of stay for emergency and elective patients separately as Supplementary Figure 3 and added a pointer to this in the results.

Line 363: This is an interesting finding, and one that has some plausibility based on patient flow through hospitals. Generally elective patients arrive for around 7am for

planned operating lists, while emergency patients may arrive at any time of day. Patients admitted in a planned fashion e.g. the night before an operation are usually clinically more complex, and thus may have harder to predict lengths of stay. Anecdotally (from my practice at least), acuity of emergency presentations varies over the course of the day, with higher acuity dominating at night (i.e. things that couldn't wait til morning). These features can be seen, particularly for elective care, in Figure 4. Perhaps consider providing explanations for the differences you see across your time comparisons, given you include them and find some variation.

Response: For elective admissions, the model performance across prediction times was very similar, with very slightly higher performance at 6 am and 12 pm. There are multiple plausible explanations, some of which are alluded to by the reviewer, but the differences were small.

There was more variation in prediction performance for emergency patients, with the best performance at 12pm and 6pm. This probably reflects in part that recent data arising from clinical reviews predominantly conducted in the morning inform these predictions. Additionally, admission of new and more unwell patients during the late afternoon and evening may make overnight and early morning predictions more challenging. We have expanded the discussion to reflect this (line 534-540).

Line 407: When discussing feature importance, it may be helpful for the reader to understand the direction of any association between the variable(s) and the outcome – e.g. does receipt of IV antibiotics in the last 24 hours increase or decrease likelihood or discharge? I also see here that the SD of historic length of stay is important. Given the likely skew of LOS in the data, could quantile measures, such as using 25th and 75th centiles be helpful here and avoid the enforced symmetry of the SD?

Response: We have added a Shapley plot to present the direction of association between the top 20 features and the outcome (discharge) as Supplementary Figure 12. The negative Shap value indicates a negative association (less likely to be discharged), while the positive value indicates a positive association (more likely to be discharged).

We agree that SD is not optimal for skewed data where the number is being interpreted directly, but it still captures variability. Given our task was prediction rather than inference, and we wanted to restrict the number of features in the dataset, e.g. rather than also adding 25th and 75th quantiles or similar for each continuous variable, we have retained the use of SD for LOS.

Line 412: How were the procedures included defined? How was the number of procedures quantified (e.g. if using OPCS codes for laparoscopic surgery may have two codes – one to describe the operation and another the approach)? Were minor and major procedures differentiated? E.g. a simple procedure has less of an impact on LOS than a major one.

Response: We included distinct OPCS codes for each procedure, and excluded modifying codes starting with 'Y' and 'Z'. We have clarified this in Supplementary Table 1. We did not differentiate between minor and major procedures because we wanted to create a generalised hospital-wide model for all patients rather than specific models for each procedure. However, we accept that it may be possible to improve performance by developing procedure or condition specific models and have now noted this as a limitation / potential future work (line 598-604).

Line 494: Different variables are likely to have different importance for different patient groups and admissions. As you are pooling a wide range of patients and types of admission, is there a chance that the dominance of common admissions reduces the predictive accuracy for rarer types of presentation? While performance plateaued after 200 features, as there further performance gains for some patient groups for including more features? It seems from the results like a major challenge is predicting when someone who has been in hospital for a long period is likely to go home. I imagine features predictive of their discharge are different to the discharge of someone who has been in hospital for 1 day.

Response: In our preliminary work we tried fitting separate models based on prior length of stay, but the models gave worse overall performance than a unified model. We agree that modelling different patient groups might select different features, but as mentioned above, we wanted to fit a unified model for all patients to be deployed in hospitals more easily. The XGB architecture does to some extent allow for different features to be differently important in different subgroups, as it fits multiple decision trees and different branches of the decision trees can be tuned to specific populations by the fitting algorithm if this improves model performance. We have expanded our discussion of this point (line 598-604).

Line 616: what does 'community' mean here?

Response: It means representatives from the public rather than from the university. We have clarified this.

Line 732: Consider expanding acronyms and abbreviations, particularly if not done in the main text already.

Response: We have added the explanations for acronyms and abbreviations in the table legend.

Line 776: These plots are great, and useful to see. In most cases, a clear difference between weekday and weekend predictions is seen. The predicted values appear markedly flatter than the actual values observed for the LR, sometimes quite markedly, but perform slightly better for the XGB. In both cases, comparing the LR and XGB plots side by side shows neither are necessarily excellent predictors of discharge numbers.

Response: Thank you for your comment. We have now included the distribution of prediction errors in Supplementary Figure 7.

Reviewer #2 (Remarks to the Author):

SUMMARY

The paper develops and validates a model for patient discharge prediction in a large UK hospital network. Overall, the analysis is thoroughly conducted, exhaustive, and clearly explained. The authors also provide a range of sensitivity analyses (using prediction times on the test set different that those used for training; analysing the impact of the size and recency of the training data on out-of-sample performance; comparing pre- and post-COVID performance) that are insightful and, as far as I can tell, not common in the literature.

I would support the publication of the manuscript upon minor revisions.

My primary concerns are:

- Misleading title. The present study does not demonstrate/support any improvement on patient flow. This is pure speculation (based on evidence from other studies). The present paper is about the development and the validation of a machine learning model for discharge prediction. It should be advertised as such.

Response: We have changed the title to: “Predicting individual patient and hospital-level discharge using machine learning.”

- Simplistic baseline. The logistic regression baseline is overly simplistic in my opinion. A more credible benchmark should be used (see detailed comments)

Response: We address this below, and in the response to reviewer 1 above.

DETAILED COMMENTS

Title "Improving patient flow through hospitals with machine learning based discharge prediction" Nothing in the study justifies the claim that patient flow "improved". This study is a retrospective analysis, evaluating the predictive power of ML models for discharge prediction. The fact that these predictions could improve patient flow is not supported in this study. Please edit the title to reflect the contribution of the present paper more faithfully

Response: We have changed the title to: “Predicting individual patient and hospital-level discharge using machine learning.”

p.2 Abstract l.36 "AUCs of 0.87" -> "AUROCs of 0.87" (since the authors use 2 types of AUC metrics)

Response: We have changed ‘AUC’ to ‘AUROC’ throughout the manuscript.

p.2 Abstract l.44 "optimising" Please check journal policy on the use of UK or US English.

Response: The use of UK English is accepted.

p.3 l.62 "e.g. prompting" -> e.g., prompting (general comment)

Response: We have changed this.

p.4 l.94 "Most studies either evaluated individual-level discharge prediction performance or hospital-wide predictions, but did not combine the two in a single approach." The term "most studies" suggests that some studies did. It would be good to have a more faithful and precise description of the literature here. Such as "As far as we know, only Study XX did ..." or "With the exception of X, Y, all of the aforementioned studies..."

Response: We have modified this to: "Apart from two studies^{11,13}, all of the aforementioned studies either evaluated individual-level discharge prediction performance or hospital-wide predictions, but did not combine the two in a single approach."

p.4 l.112-118 Exclusion criteria. Why did the authors needed to exclude both patients <= 16 yo and paediatric patients. I would think of those criteria as redundant. Are they? If not, why?

Response: Among 527,820 admissions after excluding patients aged <16y, 428 (0.08%) admissions had a consultant specialty of paediatrics, therefore we further excluded this small proportion of patients in paediatrics but aged >16y.

p.5 l.163 "because extreme gradient boosting (XGB) models can handle missing values by default" The authors should specify which package/implementation of XGB they used. Different software packages implement different strategies for dealing with missing values. See Section 5 in Josse, J., Prost, N., Scornet, E., & Varoquaux, G. (2019). On the consistency of supervised learning with missing values. arXiv preprint arXiv:1902.06931.

Response: We have added the strategies for dealing with missing values: "because extreme gradient boosting (XGB) models can handle missing values by default with a 'missing incorporated in attribute' algorithm." We have previously specified in 'Software' that we used xgboost (version 2.0.3).

p.5 l.164 "For the baseline logistic regression model, the included features (age, sex, day of the week, and hours since admission) did not have missing values." This is one of my biggest concern about the study. The baseline is overly simplistic. I recommend the authors use (a) a regularised (L1 or L2) logistic regression model;

(b) a simple imputation method (mean imputation for continuous, new category for categorical features)—note that this is not more complicated than the "average LOS for the diagnosis code" feature they are using; (c) the same set of features and overall training pipeline as their XGB model (Figure 1.d). This would be a fair comparison.

Response: As we note in our response to reviewer 1, we have now added to the comparison a regularised logistic regression model with the same 200 features as the XGB model. We have added details on this to the methods. The performance of the 200 feature LR model was higher than the simple LR model, but lower than the XGB model. The difference was bigger between the two LR models, indicating that having more carefully-designed features makes a greater difference than using a more complex model. We have updated Table 3, and added this finding to the results (line 343-349).

p.7 l.242 "are likely to apply to hospitals with similar daily discharge rates" What is the actual discharge rate in the hospital studied? It would be important to share these numbers earlier for the reader to appreciate the representativeness of the hospital

Response: The discharge rate is provided in the manuscript results: "47,177 (19.8%) and 141,531 (17.5%) discharge events within 24 hours of the index date were observed, respectively". We have clarified this is the discharge rate in the accompanying text.

p.9 l.327 "We calculated the total number of discharges expected in the next 24 hours across all elective or emergency admissions in the hospital by summing the individual-level predicted discharge probabilities." This explanation should be provided earlier, in the methods section (e.g., around p.7 l.247). In particular, with this aggregation strategy, it is important that the authors have calibrated their predicted probabilities and not use their binary predictions with the F1-optimising threshold (p.7 l.236). So, the authors should highlight these points.

Response: We have moved this explanation to the methods section, and pointed out that the predicted probabilities we used were after calibration: "For hospital-level prediction, we calculated the total number of predicted discharges expected in the next 24 hours across all elective or emergency admissions in the hospital by summing the individual-level predicted discharge probabilities after calibration. We summarised the accuracy of predictions of the total number of patients discharged using normalised mean absolute error (MAE, %), i.e. the mean of the differences in predicted and actual discharges each day (over the 365 predictions in the test dataset) divided by the mean number of discharges per day."

We have now also emphasised again in the discussion that we used calibrated probabilities.

p.9 l.363 "Sensitivity analyses by prediction time" I am a little bit confused with this

sensitivity analysis. The models have been trained to predict whether a patient, based on information on day d at noon, will be discharged on day (d+1) at noon (within 24 hours). Am I correct that the different prediction times (midnight, 6 am, 6 pm) correspond to the following prediction horizon: 12 hours, 18 hours, 30 hours? It would be good if this could be better explained in the paper. Also, does it make sense that the model performances are more robust to some prediction time than others?

Response: The different prediction times did not affect the prediction horizon, which was 24 hours. For example, predicting at midnight means predicting whether the patient was discharged from 12 am to 12 am the next day. We have clarified this in the methods.

Overall model performance was broadly consistent across prediction times, the variations likely reflect the pattern of patient flow in hospitals as discussed in our response to reviewer 1 above: “There was more variation in prediction performance for emergency patients, with the best performance at 12pm and 6pm. This probably reflects in part that recent data arising from clinical reviews predominantly conducted in the morning inform these predictions. Additionally, admission of new and more unwell patients during the late afternoon and evening may make overnight and early morning predictions more challenging. We have expanded the discussion to reflect this (line 534-540).”

p.9 l.364 "Patients were more likely to be discharged between 10 am and 8 pm," I am surprised that this observation only comes now in the study. For me, it should have informed the design of the study from the beginning. This pattern is very common and it is thus quite odd to use noon as the baseline prediction time in my opinion. Also, from an implementation perspective, noon is a very busy time in the hospital so it might be hard to properly extract data from the EHR in real-time to generate the features used for the prediction (this is one of the main reasons why deployed ML models are typically run overnight, when EHR activity is lower).

Response: For the presentation of the paper, we picked 12 pm as the main results for illustrative reasons and because this represents a time similar to that when most ward rounds are complete by. We included 6 am, 6 pm, and 12 am as sensitivity analyses. However, in reality if our models were deployed, we would envision updating the predictions throughout the day, e.g. hourly, to achieve more accurate predictions from fresh data as it is available. We have expanded the sentence in our discussion covering this (line 539-540).

p.9 l.364 "Patients were more likely to be discharged between 10 am and 8 pm," Is there some heterogeneity in that observation between elective and emergent admissions? (That could explain heterogeneity observed in line 380-385)

Response: As shown in Supplementary Figure 10, both elective and emergency patients were more likely to be discharged between 10 am and 8 pm, although emergency admissions were slightly more likely to be discharged later than elective admissions. This could be explained by the fact

that elective patients are usually admitted in the morning, while emergency patients are admitted at any time of day. It might also reflect that many elective ward rounds are completed by 9am, whereas most acute medical ward rounds last most of the morning in our setting.

We have added to our discussion some commentary on potential reasons for differences in the predictive performance of the emergency model throughout the day (discussed in more detail in our response to reviewer 1).

p.10 l.407 "Feature importance" I am surprised not to see current length of stay as an important predictive feature for elective admission. Was it included in the study? Was it selected at least as part of the top 200 features? Given the importance of that feature in other studies from the literature, the authors should comment.

Response: The current length of stay was included in the study as 'number of hours since admission'. It was selected as the top 20 important features for both elective and emergency admissions.

p.11 l.452-453 "further studies comparing model performance to clinician predictions and of trial implementations are required." The authors do not give enough credit here to studies that have done that, e.g., [7,13] or King, Z., Farrington, J., Utley, M., Kung, E., Elkhodair, S., Harris, S., ... & Crowe, S. (2022). Machine learning for real-time aggregated prediction of hospital admission for emergency patients. NPJ Digital Medicine, 5(1), 104. Na, L., Carballo, K. V., Pauphilet, J., Haddad-Sisakht, A., Kombert, D., Boisjoli-Langlois, M., ... & Bertsimas, D. (2023). Patient outcome predictions improve operations at a large hospital network. arXiv preprint arXiv:2305.15629.

Response: We have included these references and discussed the comparison between model performance and clinician predictions.

p.12 l.513 "We envision our approach could be deployed in several ways". Again, this vision is largely based on how previous studies have deployed their algorithms. Credit should be given to such studies.

Response: We have now cited examples of studies that have deployed their discharge prediction algorithms.

Reviewers' comments:

Reviewer #1 (Remarks to the Author):

Thank you for giving me the opportunity to review a revised version of this manuscript. The authors have gone to great effort to address the comments raised by the previous version and I'm satisfied with the response in the vast majority of cases. It's very useful to see the improvement in performance in the LR models through using the same set of variables as used in the XGB model. This improvement seems to be greater than the improvement resulting from using XGB over LR (though this is not explicitly calculated).

My main comment would be to ensure this finding is mentioned prominently across the manuscript, particularly in line 30 and lines 478-481. In both of these cases, the emphasis is on the difference between the baseline LR and the complete XGB models. Given the substantial improvement achieved in the LR model through simply feeding it the same data as the XGB, it think there's a danger the reader attributes the cause of the difference to the modelling method rather than the breadth of input data.

Response: We previously reported the results in line 342-348. We have now added this to the abstract and the first paragraph of the discussion to make it prominent as a main finding.

Lines 309-310: Thank you for including the supplementary figure for the distribution of LoS. I still do feel that only including the median and IQR in the main text doesn't adequately express the profound skewness of the distribution. I would recommend commending on this in the main text and linking to the figure in that context accordingly.

Response: We have added at line 310 that the length of stay was right-skewed, and most patients were discharged within a week.

I congratulate the authors on a very well-conducted study that will be of interest and use to a broad readership.

Reviewer #2 (Remarks to the Author):

All of my comments have been satisfactorily addressed by the authors.